# IntentRL: Training Proactive User-intent Agents for Open-ended Deep Research via Reinforcement Learning

Haohao Luo [1 2 *]   Zexi Li [2 *]   Yuexiang Xie [2]   Wenhao Zhang [2]   Yaliang Li [2]   Ying Shen [1 3]

## Abstract

Deep Research (DR) agents extend Large Language Models (LLMs) beyond parametric knowledge by autonomously retrieving and synthesizing evidence from large web corpora into long-form reports, enabling a long-horizon agentic paradigm. However, unlike real-time conversational assistants, DR is computationally expensive and time-consuming, creating an *autonomy-interaction dilemma*: high autonomy on ambiguous user queries often leads to prolonged execution with unsatisfactory outcomes. To address this, we propose **IntentRL**, a framework that trains proactive agents to clarify latent user intents before starting long-horizon research. To overcome the scarcity of open-ended research data, we introduce a scalable pipeline that expands a few seed samples into high-quality dialogue turns via a shallow-to-deep intent refinement graph. We further adopt a two-stage reinforcement learning (RL) strategy: Stage I applies RL on offline dialogues to efficiently learn general user-interaction behavior, while Stage II uses the trained agent and a user simulator for online rollouts to strengthen adaptation to diverse user feedback. Extensive experiments show that IntentRL significantly improves both intent hit rate and downstream task performance, outperforming the built-in clarify modules of closed-source DR agents and proactive LLM baselines.

## 1. Introduction

Large Language Models (LLMs) (Yang et al., 2025; Liu et al., 2024; Singh et al., 2025; Comanici et al., 2025) have

---
*Equal contribution [1]Sun Yat-sen University, Shenzhen, China [2]Tongyi Lab, Alibaba Group, Hangzhou, China [3]Guangdong Provincial Key Laboratory of Fire Science and Intelligent Emergency Technology, Shenzhen, China. Correspondence to: Ying Shen <sheny76@mail.sysu.edu.cn>.

*Proceedings of the 43rd International Conference on Machine Learning*, Seoul, South Korea. PMLR 306, 2026. Copyright 2026 by the author(s).

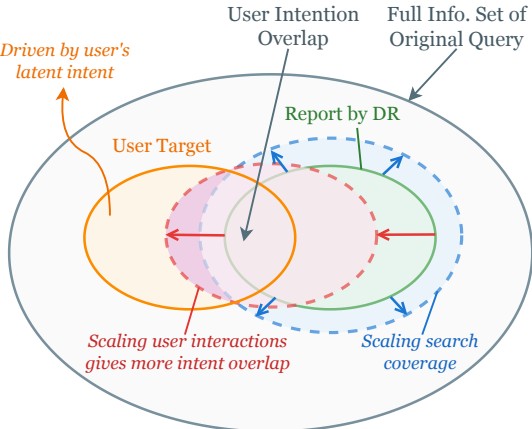

*Figure 1.* **Scaling user interactions is more efficient to reach the user's target for open-ended deep research.** The user's simple query implies a huge information set, while the underlying user intent is a subset. LLM agents will have their own implicit bias in understanding the query; without interaction, will have little overlap with the user's target. Scaling search coverage can expand the information covering but less efficient. We propose IntentRL by scaling user-agent interactions before deep research to reach better user intent alignment without increasing the deep research burdens.

demonstrated remarkable capabilities across various tasks, from mathematical reasoning (Ren et al., 2025) to comprehensive writing (Wu et al., 2024) and code generation (Cursor Team, 2025; Anthropic, 2025). Recently, Deep Research (DR) agents (Google, 2025; OpenAI, 2025a; Qwen Team, 2025) represent a significant leap in autonomous intelligence that goes beyond chatbots. DR agents autonomously reason, synthesize vast amounts of online information, and execute multi-step investigations to generate comprehensive reports for human-expert-level user tasks (OpenAI, 2025a).

However, this enhanced capability introduces a critical vulnerability: the ***autonomy-interaction dilemma***. In standard chat applications, the cost of misunderstanding a user is low since corrections can be made instantly. In contrast, DR tasks typically involve long-horizon execution (e.g., 30 minutes of browsing and reading). If an agent autonomously executes an ambiguous query without clarification, the result is a costly waste of computational resources and time, yielding a report that fails to satisfy the user.

This failure mode arises from an **information gap** between the user's explicit query and their *latent intent*. For instance, a query like "give me a report about deep research agents" may imply a broad information set (technical details such as memory, base model, agent mode; applications like finance or health care; etc.), while the user's hidden intent could be much narrower: an AI product manager who only cares about closed-source DR products. Without user-agent interaction, a DR agent is prone to follow its own implicit biases, producing an off-target report (e.g., centering on ReAct (Yao et al., 2022)).

Prior work shows that users often lack the domain knowledge or expression skills to state precise requirements (e.g., perspective or scope) (Ye et al., 2025; Qian et al., 2025a). As Figure 1 illustrates, the true intent is a subset of the query's full information set. Under limited search and time budgets, without clarification, the agent retrieves a subset shaped by its own interpretation, rather than the subset aligned with the user. Although one could try to cover the "full set" by scaling search, this is computationally intractable. A more efficient alternative is *scaling interaction*: proactively eliciting latent intents before execution to align research with the user's real needs. Current closed-source DR products (e.g., Qwen DR and OpenAI DR) typically include an intent-clarification module before the main DR task. However, our experiments show that these modules often fail to elicit sufficient user intent, and open-source, user-intent-driven DR agents remain largely unexplored.

Training proactive agents for open-ended research is hard because DR is inherently *open-ended* while high-quality data is scarce. Most proactive LLM methods target closed domains with constrained state spaces, such as medical diagnosis (fixed symptoms) (Wei et al., 2025) or tool use (finite APIs) (Wang et al., 2025b). In contrast, deep research spans unrestricted topics, making it impossible to define a "complete" information set. Meanwhile, data that reflects real-world, open-ended user intent is extremely rare: existing benchmarks (Du et al., 2025) are small (often ≈100 samples), and synthetic data may inherit LLM self-knowledge biases and miss the diversity of genuine human requests.

To address these challenges, we propose **IntentRL**, a reinforcement learning (RL) framework for training proactive agents to clarify latent user intent. We first introduce a scalable data construction pipeline: starting from a few seed samples, we build a *Clarification Directed Acyclic Graph* (C-DAG) that maps queries to both shallow (surface-level) and deep (rubric-derived) intents, then traverse it to expand seeds into a large-scale dataset of diverse clarification trajectories. We then train the agent with a two-stage RL strategy. Stage I leverages offline expert trajectories from the C-DAG via offline RL to establish a strong foundation. Stage II mitigates offline distribution shift with an online, intent-aware

user simulator, enabling exploration under feedback that penalizes redundancy and encourages adaptive clarification across diverse user responses.

Our experiments show that IntentRL yields substantial gains on downstream deep research tasks. Notably, we observe a scaling effect: the advantage of proactive clarification grows as the underlying DR agent becomes more capable. IntentRL also offers practical insights for LLM RL under non-verifiable rewards.

Our main contributions are as follows:

- We identify the autonomy-interaction dilemma in open-ended deep research and formulate latent intent clarification as a POMDP.
- We propose a data construction pipeline that scales a few seed samples into thousands of high-quality dialogue turns via a Clarification DAG, alleviating data scarcity for open-ended tasks.
- We introduce IntentRL, a two-stage reinforcement learning framework that combines offline stability with online exploration using a rule-based/LLM-judge user simulator.
- Extensive results show that IntentRL significantly improves report alignment with user needs, offering a scalable path to integrate user-centric interaction into autonomous agents. Our code is released at https://github.com/Luohh5/IntentRL.

## 2. Related Work

**Deep Research**, introduced by OpenAI for synthesizing online information into reports (OpenAI, 2025a), has evolved significantly (Huang et al., 2025b; Zhang et al., 2025). While "deep search" models like Search R1 (Jin et al., 2025), R1 researcher (Song et al., 2025), and WebSailor (Li et al., 2025a) focus on fact-based reasoning, we target open-ended report generation (Google, 2025; OpenAI, 2025a). Current evaluation in this domain lacks definitive gold standards; therefore, structured rubrics have been widely adopted by recent benchmarks, such as DeepResearch Bench (Du et al., 2025), DeepResearch Arena (Wan et al., 2025), DeepResearchGym (Coelho et al., 2025), DRBench (Abaskohi et al., 2025), Rigorous Bench (Yao et al., 2025b), LiveResearchBench (Wang et al., 2025a), and Fan et al. (2025). Methodologies rely on multi-agent workflows such as WebWeaver (Li et al., 2025b) and diffusion refinement (Han et al., 2025), while RL remains limited by the challenges of defining rewards (Shao et al., 2025). Though closed-source systems like OpenAI and Qwen Deep Research include built-in clarification modules, our studies find these limited, and open-source training practices for user-intent agents remain scarce.

**Proactive LLM** research shifts models from passive responders to active collaborators that elicit deeper user intent.

This literature focuses on (i) measurement and (ii) training. Regarding measurement, benchmarks consistently show models struggle to acquire missing information compared to solving specified prompts: AR-Bench (Zhou et al.) targets reasoning with incomplete evidence; NoisyTool-Bench (Wang et al., 2025b) and IN3 (Qian et al., 2024) emphasize ambiguous instructions; and UserBench (Qian et al., 2025a) evaluates multi-turn alignment. Regarding training, "asking" is often treated as policy learning. Ask-when-Needed (Wang et al., 2025b) and ACT (Chen et al.) improve clarification under ambiguity, while Huang et al. (2025a) rewards eliciting implicit knowledge. Furthermore, Wei et al. (2025) trains proactive consultants from offline logs, with CollabLLM (Wu et al.) and UserRL (Qian et al., 2025b) highlighting long-horizon objectives. However, most prior work targets domains with enumerable knowledge (e.g., tool use or medical consultation), rendering open-ended deep research significantly more challenging due to diverse, free-form requests that may exceed the LLM's knowledge.

## 3. Method

### 3.1. Problem Formulation

We formulate user-intent clarification as a multi-round interaction between an intent-mining agent and a user. Given an initial fuzzy query $q_s \in \mathcal{Q}_{\text{simple}}$, the agent asks a sequence of clarification questions $\mathbf{x}_{1:T} = \{x_1, \ldots, x_T\}$ and receives user clarifications $\mathbf{u}_{1:T} = \{u_1, \ldots, u_T\}$. After $T$ turns, we use a summary agent that synthesizes a fine-grained query $q_f$ from the dialogue history to provide a clearer task description for a downstream deep research agent.

We learn a policy $\pi_\theta$ of the intent-mining agent, which can proactively ask the user for clarification. The dialogue history (observation) up to turn $t$ is defined as

$$H_t \triangleq (q_s, x_1, u_1, \ldots, x_t, u_t), \quad H_0 \triangleq (q_s). \quad (1)$$

At turn $t$, the agent conditions on $H_{t-1}$ and outputs the next question

$$x_t \sim \pi_\theta(\cdot \mid H_{t-1}). \quad (2)$$

We model the interaction as a Partially Observable Markov Decision Process (POMDP) $\langle \mathcal{S}, \mathcal{A}, \Omega, \mathcal{T}, \mathcal{R} \rangle$ with: **(1) State** $\mathcal{S}$: the user's complete latent intent $I$ that is fixed throughout the dialogue and hidden from the agent, necessitating incremental inference through interactions; **(2) Observation** $\Omega$: the observable history $H_{t-1}$; **(3) Action** $\mathcal{A}$: the clarification question $x_t$ asked by the agent; **(4) Transition** $\mathcal{T}$: user response dynamics

$$u_t \sim P(\cdot \mid I, H_{t-1}, x_t), \quad (3)$$

**(5) Reward** $\mathcal{R}$: a function $R(H_{t-1}, x_t)$ that measures the value of information gained by asking $x_t$.

### 3.2. Method Overview

Training a user-intent agent faces two key challenges in the training environment: **(1)** *how to construct intent-rich dialogue data*; and **(2)** *how to model user-agent interactions in real practice*. The first requires diverse, rich, deep-research-dependent queries paired with latent intents and suitable evaluation metrics. The second requires capturing realistic interaction dynamics so the agent can learn the POMDP and respond intelligently to arbitrary user feedback.

To tackle **(1)**, we scale seed samples from DeepResearch Bench (Du et al., 2025) and construct two levels of intent: (i) *shallow intent*, by fuzzifying the original query into a simpler one and treating the resulting information gap as intent; and (ii) *deep intent*, by defining the user-demand gap between the original query and the rubric requirements. We then build a clarification DAG from these intent lists, and enlarge it by expanding alternative intent options. To tackle **(2)**, we combine offline and online reinforcement learning. We generate offline dialogues by traversing the DAG, and build an online user simulator via embedding-based thresholding and intent-aware prompting. Training proceeds in two stages: *Stage I* uses offline dialogues to learn general interaction behavior efficiently, and *Stage II* uses the Stage-I agent with the simulator for online rollouts, reinforcing proactive and adaptive responses across diverse user reactions.

### 3.3. Data Construction: Scaling Intent Data from Seed Samples

**Motivation** Deep research reports are evaluated by rubrics that define a "gold standard" for comprehensiveness, insight, and related criteria. In practice, there is often a large information gap between the user's query and what the rubrics require. Some gaps can be filled by external retrieval, but many reflect latent user preferences about scope, focus, and depth. For example, a user may implicitly expect specific analytical perspectives (e.g., market structure, key challenges, emerging trends) that are covered by the rubrics but not stated in the query. By eliciting such preferences through targeted clarifications before research begins, the agent can align its investigation with the rubrics from the outset, avoid costly detours, and improve efficiency. We therefore ground clarifications in rubric-derived intents, teaching the agent to bridge query-rubric gaps via interaction. The data construction process is mostly empowered by LLMs.

**Intent Construction** *Shallow Intent:* DeepResearch Bench provides detailed task queries, but real users often ask ambiguous questions (Ye et al., 2025). To model this, we derive a simple query from each original one. Starting from the original query $\mathcal{Q}_{\text{origin}}$, we identify and remove explicit constraints to obtain a fuzzy query $q_s$. The removed constraints are preserved as *shallow intents* $I_s$. *Deep Intent:* We further model rubric-derived deep intents. We

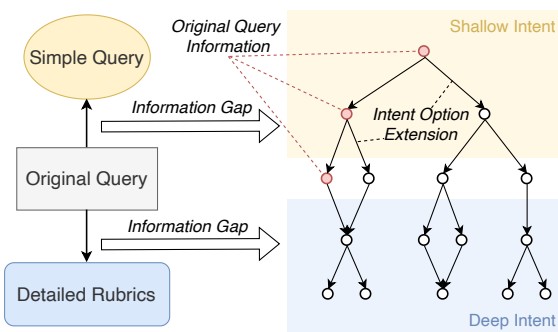

*Figure 2.* **Demonstration of C-DAG construction.** We derive a simple query from the original and extract the removed constraints as shallow intents. We then derive deep intents from the gap between the original query and the rubrics. Finally, we extend the C-DAG by adding additional intent options to question edges.

analyze the rubric set $\mathcal{C}$ to identify requirements that cannot be met by retrieval alone, but instead depend on the user's preferred focus or perspective. We formalize these preference-dependent requirements as *deep intents* $I_d$.

**Clarification Graph (C-DAG) Construction**    To scale data, we propose a *Rubrics-to-Clarify* pipeline that converts static scoring rubrics into a *Clarification Directed Acyclic Graph* (C-DAG) $\mathcal{G} = (\mathcal{V}, \mathcal{E})$, illustrated in Figure 2. We define the overall latent intent as $I \triangleq \{I_s, I_d\}$. Conditioned on $I$, we generate clarification questions with multiple options (including rubric-derived alternatives) and build the C-DAG in two steps: (i) construct a base graph from shallow intents to cover missing surface constraints; (ii) expand it with deep intents to increase analytical depth and breadth. Each node $v \in \mathcal{V}$ represents a clarification question (with multiple options), and each directed edge $(v_i, v_j) \in \mathcal{E}$ encodes a logical dependency: $v_j$ becomes meaningful only after resolving $v_i$. This structure enables systematic generation of diverse clarification trajectories spanning both shallow constraints and deep perspectives.

**Trajectory Generation via Graph Traversal**    We generate a large dataset $D$ of diverse yet coherent clarification trajectories via depth-first traversal of the C-DAG from its root nodes. We maintain a stack $\mathcal{K}$ of currently visitable nodes $\mathcal{V}^\star$. At each step, we pop and visit a node. When a question offers multiple options, we branch the traversal by instantiating distinct paths for different option selections, scaling a single benchmark task into many intent-aligned trajectories.

### 3.4. Agent Interaction Modeling: from Offline to Online

**Offline User-Agent Interaction Dialogues**    From the clarification trajectories generated by C-DAG traversal, we obtain an intent list ordered by intent depth. We then prompt a strong LLM to mimic user-agent interaction and generate multi-turn dialogues. Starting from 50 seed samples, we expand to 371 intent trajectories, yielding 2347 dialogue

turns $\{x, u\}$. These dialogues serve as expert trajectories, containing ground-truth questions and ideal user responses, and are used in Stage I of RL training.

**Online Intent-Aware User Simulator**    Offline RL can suffer from distribution shift in real-world settings, especially with open-ended and diverse action/state spaces. We therefore introduce an online, intent-aware user simulator for RL rollouts and evaluation. The simulator is initialized with a predefined latent intent set $\mathcal{I}_{\text{latent}}$, and all responses are strictly grounded in these ground-truth intents. Concretely, it uses a hybrid *rule-based* and *LLM-judge* design: (1) *Rule-based logic:* it measures semantic overlap with the `all-mpnet-base-v2` embedding model. If the similarity between the agent's current question and the interaction history exceeds a repetition threshold $\tau_{rep}$, the simulator flags redundancy; if the similarity to the intent set falls below a relevance threshold $\tau_{rel}$, it flags the question as unimportant or irrelevant. (2) *LLM response:* only if the question passes these checks does an LLM generate a user response conditioned on the intent list. Interaction runs for a fixed number of turns $T$. After $T$ rounds, a summary agent aggregates the final history $H_T$ and synthesizes a refined query $q_f$ for downstream research.

### 3.5. Two-Stage Reinforcement Learning

We propose a two-stage training framework that leverages offline expert trajectories and online simulation to achieve both stability and exploration. We use Group Relative Policy Optimization (GRPO) (Shao et al., 2024) as the RL algorithm in both stages.

**Stage I: Offline RL with Static Trajectories**    Following Wei et al. (2025), we adopt a hindsight-driven bootstrapping strategy that converts long-horizon learning into turn-level optimization by leveraging the observed future of expert trajectories. Concretely, for each turn $t$ we define a *Target Intent Set* $\mathcal{I}_t^\star$ representing the latent intents that the agent should aim to elicit at that turn.

We ground $\mathcal{I}_t^\star$ in the C-DAG traversal state by mapping it to the set of currently accessible nodes at the top of the traversal stack $\mathcal{K}$:

$$\mathcal{V}_t^\star \triangleq \text{Top}(\mathcal{K}_t), \qquad \mathcal{I}_t^\star \triangleq \{ I(v) : v \in \mathcal{V}_t^\star \}, \quad (4)$$

where $I(v)$ denotes the intent associated with node $v$ (e.g., its semantic description), and $\mathcal{K}_t$ is the traversal stack induced by the expert trajectory prefix up to $t-1$.

We train a base policy $\pi_\theta^I$ to ask questions that maximize an information-based reward aligned with $\mathcal{I}_t^\star$. Using the offline histories from $D$, we optimize the turn-level objective

$$\max_\theta J^{(I)}(\theta) \triangleq \mathbb{E}_{(H_{t-1}, \mathcal{I}_t^\star) \sim D} \\ \left[ \mathbb{E}_{x_t \sim \pi_\theta^I(\cdot | H_{t-1})} \big[ R(H_{t-1}, x_t; \mathcal{I}_t^\star) \big] \right], \quad (5)$$

where $R(H_{t-1}, x_t; \mathcal{I}_t^\star)$ is defined in the later reward formulation section. This training encourages a goal-oriented strategy that prioritizes high-value questions aligned with the intended clarifications.

**Stage II: Online Refinement with Simulated Rollouts** The policy $\pi_\theta^I$ learned from static trajectories can be brittle under its own induced distribution. In Stage I, we observe repeated behaviors, irrelevant questions, and stubborn reactions to user feedback. We therefore introduce Stage II to augment offline data with on-policy simulated rollouts.

Specifically, for each simple query $q_s$ in the Stage-I training split $D_{\text{train}}^{(I)}$, we sample $n$ rollouts using $\pi_\theta^I$. To expose the policy to different conversation depths, we progressively construct rollouts by seeding the context with ground-truth prefixes of varying lengths (i.e., partial teacher forcing), and then letting $\pi_\theta^I$ continue the dialogue. An intent-aware user simulator, initialized with the ground-truth latent intent set, responds to each agent question. This produces simulated trajectories that we decompose into turn-level samples and mix with Stage-I data to form a new training set $D_{\text{train}}^{(II)}$.

We then train a refined policy $\pi_\theta^{II}$ with the same optimization objective as Stage I, while adding rewards that penalize repetitive and irrelevant actions to encourage intelligent adaptation. By combining expert-trajectory diversity with the distribution matching of online rollouts, this hybrid training improves robustness.

**Reward Formulation** We define a turn-level reward $R(H_{t-1}, x_t; \mathcal{I}_t^\star)$ using three components: a content score, a format score, and a penalty term. The $\beta > 0$ term is a tunable knob balancing the preference for content quality and format compliance, and we set $\beta = 2$. Let $\mathcal{P}(H_{t-1}, x_t) \geq 0$ denote the total penalty magnitude. The reward is

$$R(H_{t-1}, x_t; \mathcal{I}_t^\star) =$$
$$\begin{cases} \beta \cdot R_{\text{con}}(x_t, \mathcal{I}_t^\star) + R_{\text{fmt}}(x_t), & \text{if } \mathcal{P}(H_{t-1}, x_t) = 0, \\ -\beta \cdot \mathcal{P}(H_{t-1}, x_t) + R_{\text{fmt}}(x_t), & \text{if } \mathcal{P}(H_{t-1}, x_t) > 0. \end{cases} \quad (6)$$

- **Content Score** ($R_{\text{con}}$). To measure whether a question is informative, we compare it against the target intent set $\mathcal{I}_t^\star$. Using a sentence embedding model $\phi(\cdot)$, we compute the maximum cosine similarity between the generated question and any target intent:

$$R_{\text{con}}(x_t, \mathcal{I}_t^\star) = \max_{i \in \mathcal{I}_t^\star} \frac{\phi(x_t) \cdot \phi(i)}{\|\phi(x_t)\| \, \|\phi(i)\|}. \quad (7)$$

The content score is effective only if the agent's behaviors don't fall into the penalties. We compute this on question rather than user's answer to ensure more precise credit assignment since answer rewarding entangles with LLM linguistic randomness, while avoiding variability introduced by answer generation.

- **Format Score** ($R_{\text{fmt}}$). To encourage concise and user-friendly questions, we define $N_q(x_t)$ as the number of distinct sub-questions contained in $x_t$ and set

$$R_{\text{fmt}}(x_t) = \begin{cases} 1, & \text{if } N_q(x_t) = 1, \\ 0.5, & \text{if } N_q(x_t) = 2, \\ 0, & \text{if } N_q(x_t) > 2. \end{cases} \quad (8)$$

Ideally, the agent needs to ask one proper question at a time, where we set the full format score. It is acceptable when two questions are asked (half score), but more is not.

- **Penalty Mechanism** ($\mathcal{P}$). We introduce penalty terms to suppress undesirable behaviors, including repetition, insignificance, and deviation:

$$\mathcal{P}(H_{t-1}, x_t) \triangleq \mathcal{P}_{\text{rep}} + \mathcal{P}_{\text{inv}} + \mathcal{P}_{\text{div}}. \quad (9)$$

In our implementation, repetition and deviation penalties will be triggered only if the generated question is judged as redundant or irrelevant. They are calculated based on counts accumulated in the history:

$$\mathcal{P}_{\text{rep}} = \gamma \cdot (C_{\text{rep}} + 1), \qquad \mathcal{P}_{\text{div}} = C_{\text{div}}, \quad (10)$$

where $C_{\text{rep}}$ and $C_{\text{div}}$ denote the number of redundant and deviant questions detected in $H_{t-1}$, respectively. We leave $\mathcal{P}_{\text{inv}}$ as a detector that penalizes questions that merely restate $q_s$ without gaining new information. The $\gamma > 0$ controls the severity of the repetition penalty (we set $\gamma = 2$) and the $+1$ offset ensures a non-zero penalty is applied even upon the first instance of a repetition. It is notable that the penalty scores are only applied to the Stage II training, since Stage I is the expert trajectory without flaws, whereas Stage II has free exploration. The format score and insignificance penalty are judged by an LLM, while the repetition and deviation penalties are calculated based on the sentence embedding model.

## 4. Experiments

### 4.1. Experiment Setup

**Training** For the seed samples, we randomly select half subset (50 samples) of DeepResearch Bench (Du et al., 2025) with the same distributions as the full set. We train a `Qwen2.5-7B` model as our proactive agent, with both Stage I and Stage II initializing independently from the original pre-trained checkpoint rather than being trained sequentially. More details can be found in Appendix A.1.

**Evaluation** We evaluate IntentRL at two levels. First, we measure clarification quality in both offline and online settings. Second, we place our proactive agent upstream of off-the-shelf DR agents and evaluate the resulting research reports, testing whether pre-research clarification improves intent alignment during the research

*Table 1.* Evaluation results of clarification generation on DeepResearch Bench. Intent precision and recall are reported in %.

| Method | Quality Score | Intent Prec. | Intent Recall |
|--------|--------------|--------------|---------------|
| Qwen-DR | - | 9.25 | 6.03 |
| OpenAI-DR | - | 11.32 | 6.96 |
| Learn-to-Ask | 2.28 | 21.11 | 18.45 |
| Tell Me More | 1.44 | 5.19 | 5.62 |
| CollabLLM | 2.15 | 18.00 | 12.68 |
| **IntentRL** | **2.43** | **36.44** | **27.49** |

process and ultimately enhances report quality. We use `Qwen2.5-72B-Instruct` as the summary agent, summarizing interaction history into a fine-grained query for downstream DR. ***Benchmarks:*** We use the remaining 50 samples in DeepResearch Bench (Du et al., 2025) as the test set. To assess generalization, we further evaluate on two disjoint benchmarks: Rigorous Bench (Yao et al., 2025a) and Personalized Deep Research (PDR-Bench) (Liang et al., 2025). Rigorous Bench contains 214 expert-curated DR tasks from domains largely different from DeepResearch Bench; we randomly sample 50 tasks and apply our query fuzzification and intent extraction to construct evaluation data. PDR-Bench evaluates personalization by pairing 50 real-world DR tasks across 10 domains with 25 authentic user profiles. Since it provides user profiles directly, we use the original task–user pairings without query simplification, randomly sampling users to form 50 personalized instances.

**Metrics** For clarification quality, we use three metrics: *Quality Score*, *Intent Precision*, and *Intent Recall*. Quality Score is computed as $2R_{con} + R_{fmt}$ on the offline dialogue test set. For online interaction, Intent Precision and Intent Recall measure, respectively, the proportion of all ground-truth intents covered by its questions and the fraction of the agent's questions that match ground-truth intents.

For downstream report quality, we select benchmark-specific metrics most indicative of intent alignment and most likely to benefit from clarification. On DeepResearch Bench, we report *RACE* (Comp., Insight, Instr., Read., and an overall weighted score). On Rigorous Bench, we use *Semantic Quality (Quality)* for general quality assessment and *Semantic Drift (SDrift)* to quantify thematic deviation. The SDrift metric is transformed into a positive scoring factor via $1 - $ SDrift. On PDR-Bench, we report *Personalization Alignment (P-Score)* and *Content Quality (Q-Score)*. These choices focus evaluation on aspects where clarification should yield the largest gains. More detailed sub-dimensions metrics in PDR-Bench are shown in Appendix B.1.

**Downstream DR Agents and Baselines** We run end-to-end experiments with four state-of-the-art DR agents: Qwen Deep Research (Qwen Team, 2025), Gemini Deep Research (Google, 2025), OpenAI Deep Research (OpenAI, 2025a), and Tavily Deep Research (Tavily, 2025). We compare against the built-in clarification modules in Qwen

Deep Research (Qwen Team, 2025) and OpenAI Deep Research (OpenAI, 2025a), as well as proactive LLM baselines including CollabLLM (Wu et al.), Tell Me More (Qian et al., 2024), and Learn-to-Ask (Wang et al., 2025b). Notably, the built-in modules are specific to their own DR agents, whereas the proactive LLM baselines can be applied across all DR agents. **Note:** Please refer to the appendix for more implementation details, e.g., Appendix A for data and training setups and Appendix F for used prompts.

### 4.2. Main Results

**Evaluation Results of Clarification Generation** Table 1 directly evaluates the clarification questions produced by all methods, both on offline test sets and in online interactive settings with user simulators. Across all three metrics, IntentRL *consistently* outperforms every baseline.

On the offline test set, IntentRL achieves the highest Quality Score, indicating that our training pipeline learns a robust policy for selecting *context-optimal* follow-up questions that elicit the most valuable missing information. In online interaction with simulators, IntentRL attains the highest Intent Precision, showing that it uncovers more true latent intents within limited turns, and adapts its strategy based on user feedback to avoid common failures such as repetitive questioning and topic drift. Meanwhile, the best Intent Recall further confirms that IntentRL reliably captures missing information that is critical to the user's underlying research goal.

Although baselines such as Learn-to-Ask and CollabLLM perform reasonably well, they still lag behind IntentRL by a clear margin, especially in Intent Precision (-15.33) and Intent Recall (-9.04). This gap suggests that open-ended intent clarification requires both intent-rich supervision and on-policy interaction refinement, rather than prompt-level or domain-restricted training alone.

**Evaluation Results of Report Generation** Table 2 reports downstream report-generation results. IntentRL *consistently* improves report quality, outperforming all proactive LLM baselines and, in most metrics, even the built-in Clarify Modules of closed-source DR agents, achieving state-of-the-art overall performance. Among baselines, Learn-to-Ask and Tell Me More, which are specialized for ambiguity handling, typically outperform built-in modules. CollabLLM leverages its online RL framework with multi-turn-aware rewards and yields the best baseline scores, yet still trails IntentRL. This result supports the effectiveness of our data augmentation and training design.

Notably, while gains in Readability are small, we observe substantial improvements in Comprehensiveness, Insight/Depth, and Instruction-Following. This pattern indicates that clarification primarily boosts report quality and task adherence by eliciting both shallow constraints and

*Table 2.* Evaluation results of downstream deep research report generation. Best in **Bold** and second best in Underline.

| Method | DeepResearch Bench | | | | | Rigorous Bench | | PDR-Bench | |
|---|---|---|---|---|---|---|---|---|---|
| | Comp. | Insight | Instr. | Read. | Overall | Quality | 1 − SDrift | P-Score | Q-Score |
| **Qwen Deep Research Agent** | | | | | | | | | |
| No clarify | 36.92 | 39.31 | 37.27 | 46.32 | 39.39 | 53.42 | 58.37 | 5.73 | 6.71 |
| + Qwen clarify | 38.60 | 40.66 | 39.07 | 46.48 | 40.72 | 54.68 | 59.93 | 4.59 | 5.29 |
| + Learn-to-Ask clarify | 39.62 | 41.87 | 40.51 | 46.88 | 41.81 | 53.40 | 58.37 | 6.08 | 6.81 |
| + Tell Me More clarify | 36.47 | 39.22 | 37.08 | 46.65 | 39.32 | 54.15 | 58.79 | 6.02 | 6.73 |
| + CollabLLM clarify | 39.65 | 42.16 | 40.40 | 46.83 | 41.86 | 54.95 | 59.72 | 6.11 | 6.92 |
| **+ IntentRL clarify** | **41.49** | **44.06** | **43.19** | **47.04** | **43.65** | **55.24** | **60.01** | **6.17** | **6.99** |
| **Gemini Deep Research Agent** | | | | | | | | | |
| No clarify | 36.66 | 38.17 | 37.74 | 48.06 | 39.41 | 60.50 | 62.47 | 6.52 | 7.19 |
| + Learn-to-Ask clarify | 40.87 | 43.43 | 43.07 | **49.42** | 43.66 | 60.59 | 62.47 | 6.79 | 7.42 |
| + Tell Me More clarify | 37.44 | 40.42 | 38.45 | 48.24 | 40.56 | 61.51 | 63.20 | 6.84 | 7.24 |
| + CollabLLM clarify | 40.37 | 43.28 | 42.11 | 48.83 | 43.13 | **62.57** | **64.38** | 6.83 | 7.30 |
| **+ IntentRL clarify** | **43.10** | **45.02** | **44.04** | 48.50 | **45.27** | 62.47 | 63.51 | **7.06** | **7.48** |
| **OpenAI Deep Research Agent** | | | | | | | | | |
| No clarify | 30.46 | 30.50 | 34.62 | 48.42 | 36.62 | 52.79 | 58.64 | 5.29 | 6.26 |
| + OpenAI clarify | 37.82 | 36.87 | 40.09 | 50.76 | 40.26 | 52.58 | 59.36 | 5.99 | 6.36 |
| + Learn-to-Ask clarify | 36.24 | 36.95 | 40.61 | 50.48 | 41.50 | 52.80 | 58.69 | 5.85 | 6.51 |
| + Tell Me More clarify | 31.47 | 32.80 | 34.10 | 48.08 | 36.99 | 54.70 | 61.26 | 6.00 | 6.39 |
| + CollabLLM clarify | 35.87 | 37.28 | 38.46 | 50.05 | 40.12 | **55.93** | **62.90** | 6.05 | 6.54 |
| **+ IntentRL clarify** | **40.71** | **40.07** | **42.71** | **50.78** | **44.86** | 55.70 | 61.87 | **6.10** | **6.55** |
| **Tavily Deep Research Agent** | | | | | | | | | |
| No clarify | 37.12 | 39.10 | 37.78 | 45.64 | 39.42 | 60.31 | 55.76 | 6.62 | 7.28 |
| + Learn-to-Ask clarify | 39.16 | 41.08 | 40.31 | **45.99** | 41.12 | 59.98 | 55.75 | 6.91 | 7.48 |
| + Tell Me More clarify | 36.21 | 37.95 | 36.61 | 45.50 | 38.58 | 60.21 | 56.06 | 7.01 | 7.48 |
| + CollabLLM clarify | 38.20 | 40.23 | 39.52 | 45.17 | 40.39 | 61.62 | **57.15** | 7.12 | 7.60 |
| **+ IntentRL clarify** | **39.90** | **42.17** | **41.98** | 45.39 | **42.14** | **62.05** | 56.93 | **7.21** | **7.64** |

*Table 3.* Ablation study results of clarification generation and report generation on DeepResearch Bench.

| Method | Clarification Generation | | | Report Generation | | | | |
|---|---|---|---|---|---|---|---|---|
| | Clarify Quality | Intent Prec. | Intent Recall | Comp. | Insight | Instr. | Read. | Overall |
| **IntentRL** | **2.43** | **36.44** | **27.49** | **41.49** | **44.06** | **43.19** | 47.04 | **43.65** |
| w/o C-DAG Data Scaling | 2.29 | 5.78 | 5.28 | 36.94 | 39.60 | 37.69 | 46.46 | 39.61 |
| w/o Embedding Similarity Reward | 2.10 | 3.56 | 5.22 | 36.79 | 39.63 | 37.75 | 46.83 | 39.65 |
| w/o Penalty Mechanism | 2.43 | 12.67 | 15.78 | 39.70 | 42.05 | 40.80 | 47.08 | 41.96 |
| w/o Stage II Training | 2.43 | 22.89 | 20.30 | 40.65 | 43.09 | 41.97 | **47.14** | 42.89 |
| w/o Stage I Training | 1.65 | 16.22 | 15.97 | 37.08 | 39.58 | 38.50 | 46.36 | 39.81 |

deep, rubric-level intent.

To test generalization and reduce overfitting concerns, we further evaluate on Rigorous Bench and PDR-Bench. On Rigorous Bench, most proactive baselines provide only marginal gains over the no-clarify setting, and some underperform built-in clarification modules. Qualitatively, many Rigorous Bench tasks demand up-to-date knowledge (e.g., analyses of recent model releases beyond typical knowledge cutoffs), which makes it harder for proactive agents to identify what is ambiguous and worth clarifying. In contrast, results on PDR-Bench demonstrate that appropriate clarification yields *large* improvements in both intent alignment and report quality. Moreover, IntentRL remains the top performer even when using the original task–user pairings, suggesting that our data construction is *not* tailored with biases and that the learned policy is robust and generalizable

across diverse tasks and user personas. Additional results on original (non-fuzzified) queries from DeepResearch Bench and Rigorous Bench are provided in Appendix B.2.

Finally, these results show that our proactive agent can be plugged into a wide range of downstream DR agents and *reliably* improves performance, highlighting practical value. The associated overhead of our IntentRL is modest: it is trained from only 50 seed samples, and fine-tuning a 7B model is a one-time upfront cost. This investment is justified because executing an ambiguous query without clarification can waste substantial computational resources during long-horizon Deep Research tasks, whereas generating a few clarification turns is negligible by comparison. The cost-benefit trade-off therefore favors proactive clarification. We also observe a clear scaling trend: DR agents with stronger intrinsic search and reasoning benefit more from clarifica-

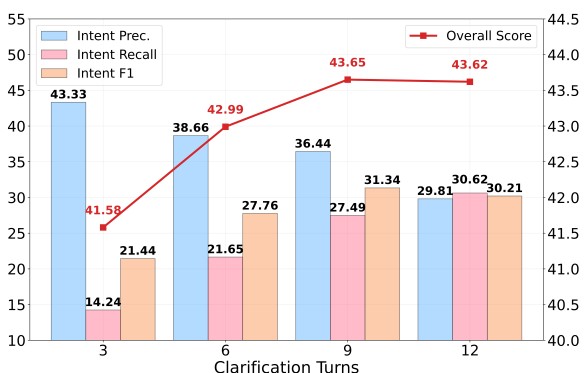

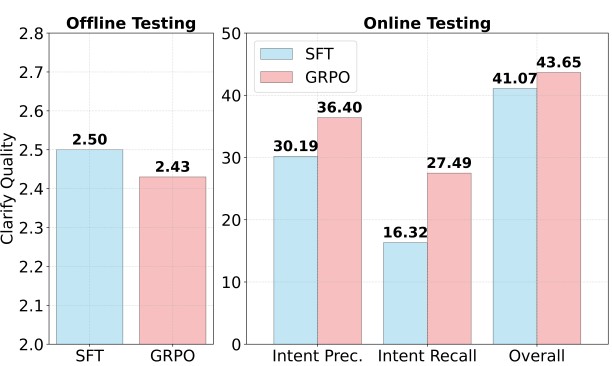

*Figure 3.* The clarification quality (%) and report overall performance with varying number of clarification turns on DeepResearch Bench.

*Figure 4.* The performance of utilizing different training algorithms in both offline and online settings on DeepResearch Bench.

tion. *IntentRL is therefore especially effective at unlocking advanced DR agents by aligning their capabilities with the user's true intent.*

### 4.3. Ablation Study

To understand which components drive IntentRL's performance, we run ablations on DeepResearch Bench that target data construction, the training framework, and reward design, as shown in Table 3. We summarize several key findings below:

- **Data Construction**: When we remove C-DAG traversal and train only on the original DeepResearch Bench data (with the *Target Intent Set* $\mathcal{I}_t^\star$ grounded as all nodes in the graph), performance drops substantially in both clarification generation and downstream report quality. This shows that our C-DAG construction and traversal not only scales high-quality training data, but also teaches the agent a more effective hierarchical organization of questions for multi-turn clarification.
- **Reward Mechanisms**: Replacing the sentence-embedding similarity with an LLM-as-judge to compute the content score degrades performance across all metrics in both offline and online settings, suggesting that embedding similarity provides a more stable and suitable objective for optimizing clarification quality. In addition, removing the penalty term has little effect offline but causes a clear drop online as the agent becomes less responsive to user feedback and produces more repetitive or off-topic questions. This highlights the penalty's importance for adaptive, user-aware interaction.
- **Training Framework**: Removing Stage II largely preserves offline clarification quality, but significantly hurts online performance, indicating that offline-only training suffers from exposure bias and does not transfer well to interactive dynamics. Conversely, removing Stage I and relying only on the pretrained model leads to a severe collapse, underscoring the necessity of our structured training pipeline for learning effective clarification strate-

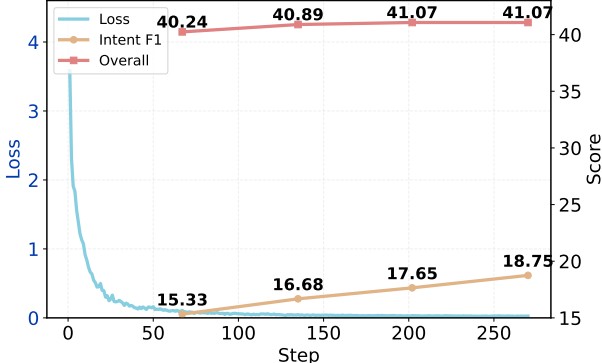

*Figure 5.* Stage II SFT training loss curve (left axis) and per-epoch clarification quality and report overall performance (right axis).

gies. Additional qualitative comparisons between stages are provided in Appendix C.

### 4.4. Further Analysis

**Effect of the Number of Clarification Turns** We study the maximum number of clarification turns $T$ to balance user experience and intent capture, as shown in Figure 3. As $T$ increases from 3 to 9, Intent Precision decreases from 43.33 to 36.44 while Intent Recall increases from 14.24 to 27.49: the agent uncovers more latent intents, but a smaller fraction of questions are directly intent-matching. To quantify this trade-off, we report Intent F1, which rises steadily from 21.44 to 31.34 over the same range. Together with the improving overall report score, this suggests that longer clarification up to 9 turns yields more useful intent coverage that benefits downstream research. When $T$ increases further from 9 to 12, the trend reverses: Recall improves only marginally, while both Intent F1 and the overall report score decline. This indicates diminishing returns and added noise from overly long dialogues. We therefore set $T = 9$ as the best-performing choice in our experiments.

**Effect of Different Training Algorithms** We evaluate supervised fine-tuning (SFT) against Group Relative Policy

*Table 4.* The performance comparison of SFT+GRPO hybrid with our GRPO training pipeline on DeepResearch Bench.

| Strategy | Comp. | Insight | Instr. | Read. | Overall |
|---|---|---|---|---|---|
| SFT+GRPO | 41.34 | 43.46 | 42.12 | **47.17** | 43.06 |
| Ours | **41.49** | **44.06** | **43.19** | 47.04 | **43.65** |

*Table 5.* The performance of utilizing different summary agents.

| Model | Comp. | Insight | Instr. | Read. | Overall |
|---|---|---|---|---|---|
| No summary | **41.73** | 43.96 | **43.24** | 45.79 | 43.51 |
| Qwen2.5-7B | 41.37 | **44.06** | 43.12 | 46.37 | 43.53 |
| Qwen2.5-72B | 41.49 | **44.06** | 43.19 | **47.04** | **43.65** |

Optimization (GRPO) using our fixed two-stage pipeline. For Stage II online SFT, a partial teacher-forcing mechanism seeds contexts with ground-truth prefixes to simulate one-turn interactions, explicitly teaching missed intent recovery. Figure 4 demonstrates SFT achieves higher offline Clarify Quality but poorer online intent capture and downstream report quality. This suggests that by optimizing next-token prediction, SFT memorizes specific offline paths rather than learning set coverage, reducing robustness to distribution shifts and real-time interaction. To rule out overfitting, we monitor SFT training and apply early stopping to choose the best checkpoint by validation performance. As shown in Figure 5, training loss steadily decreases, while both Intent F1 and Overall scores improve across epochs without degradation, confirming that the gap is due to inherent exposure bias rather than overfitting.

Furthermore, Table 4 shows a hybrid Stage I SFT and Stage II GRPO paradigm underperforms our unified GRPO strategy (yielding an Overall score of 43.06 versus 43.65). Our framework requires goal-oriented exploration grounded in C-DAG traversal states. Stage I SFT compels offline path memorization, whereas Stage I GRPO leverages explicit exploration incentives and hindsight-driven strategies to discover and cover remaining intents from any given state.

**Effect of Summary Agent**  We further analyze whether downstream gains of IntentRL depend on the summary agent converting dialogues into refined queries. In our main experiments, we use Qwen2.5-72B-Instruct as the summary agent. To isolate its contribution, we conduct an ablation on DeepResearch Bench by replacing it with Qwen2.5-7B-Instruct, and by removing summarization entirely (directly passing the raw clarification dialogue). Table 5 indicates downstream report quality remains stable across settings, with overall scores dropping marginally from 43.65 to 43.53 and 43.51. This suggests that the gains of IntentRL primarily stem from uncovering latent user constraints via high-quality intent clarification rather than powerful summarization models. Nevertheless, we recommend utilizing a capable model when feasible for summarization to construct a concise, well-structured query that minimizes context noise for the downstream DR agent.

*Table 6.* Downstream report overall scores on unseen domains when trained exclusively on Science & Technology data.

| Method | Literature | Education & Jobs | Health |
|---|---|---|---|
| No clarify | 34.82 | 42.85 | 45.02 |
| Qwen clarify | 41.38 | 44.56 | 46.70 |
| CollabLLM | 36.75 | 43.55 | 46.47 |
| IntentRL | **47.34** | **47.41** | **49.51** |

**Analysis of Generalization and Scalability**  As described in Section 4.2, IntentRL achieves top performance on personalization dimensions in PDR-Bench, which contains authentic profiles and real-world tasks, indicating successful generalization to real-world scenarios. To validate this further, we conduct an out-of-domain experiment. We trained the proactive agent exclusively on Science & Technology instances from DeepResearch Bench and evaluated it on unseen domains: Literature, Education & Jobs, and Health. As shown in Table 6, IntentRL consistently outperforms the no-clarify baseline, the training-free Qwen clarify module, and the strongest trained baseline CollabLLM across all three domains. This confirms our framework learns transferable clarification behaviors rather than domain-specific templates, proving that our improvements in out-of-domain scenarios are highly effective.

Regarding scalability, while rubrics are required as seeds in our framework, we consider this practical since 1) many Deep Research benchmarks provide them, 2) LLMs generate them with minimal oversight, and 3) our cross-domain results demonstrate learned clarification strategies transfer to unseen tasks without data reconstruction.

**More Results**  Due to space limit, we leave additional results in the appendix. Please refer to Appendix B for results on original queries of benchmarks (DeepResearch Bench and Rigorous Bench) and detailed results on PDR-Bench. And a case study of IntentRL's interaction trajectories is presented in Appendix C.

## 5. Conclusion

Deep Research (DR) faces an autonomy-interaction dilemma, where query ambiguity risks costly misalignment in long-horizon tasks. We address this with IntentRL, which formulates clarification as a POMDP to resolve latent intent prior to execution, shifting focus from scaling search to scaling interaction. We introduce a data pipeline for trajectory generation and a two-stage RL scheme that combines offline expert learning with an intent-aware simulator to minimize redundancy. Experiments demonstrate that IntentRL consistently outperforms built-in modules and baselines in intent capture and report quality, with gains scaling on stronger models. We hope these findings offer a practical foundation for integrating user-centric interaction into future long-horizon autonomous systems.

## Acknowledgements

This work was supported in part by the New Generation Artificial Intelligence-National Science and Technology Major Project (2025ZD0123003), the National Natural Science Foundation of China Enterprise Innovation and Development Joint Fund (Artificial Intelligence Field) Key Support Projects (U25B2072).

## Impact Statement

This paper studies how to improve user–agent alignment in Deep Research (DR) systems by training proactive intent-clarification agents. If deployed responsibly, the proposed approach can reduce wasted computation and user time by preventing long-horizon research from pursuing the wrong scope, and can improve accessibility by helping users who struggle to specify requirements articulate their needs through guided interaction. The work may also benefit open research by providing an open-source pathway for intent-driven DR agents, rather than relying solely on proprietary clarification modules. Furthermore, this paper presents work whose goal is to advance the field of Machine Learning. There are many potential societal consequences of our work, none which we feel must be specifically highlighted here.

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

# A. More Implementation Details

## A.1. Data Construction

*Table 7.* Statistics of clarification datasets created through C-DAG based on DeepResearch Bench.

|   | # Train | # Graph | Max # Depth | Avg # Depth | Min # Depth | Max # Trajectory | Avg # Trajectory | Min # Trajectory |
|---|---------|---------|-------------|-------------|-------------|------------------|------------------|------------------|
| # | 50 | 50 | 8 | 4.4 | 2 | 45 | 7.4 | 1 |

We mainly utilize Large Language Model to construct and scale intent data from seed samples. We first employ `gpt-4.1-2025-04-14` to simplify the original query, extract the shallow intents and model the rubric-derived deep intents. Subsequently, we utilize `gemini-3-pro-preview` to build a base graph and expand it by recursively traversing and branching each node based on the shallow and deep intents. We then employ a Depth-first Search (DFS) strategy to generate multiple clarification trajectories for each graph. Finally, we also use `gemini-3-pro-preview` to simulate both proactive agents and users to generate dialogues that are more authentic and better aligned with real-world scenarios based on the trajectories. Through this data construction pipeline, we successfully scale the original 50 seed samples from DeepResearch Bench into nearly ×50 high-quality dialogue instances for training. Table 7 presents the detailed statistics of the constructed dataset. Specifically, we construct one C-DAG for each of the 50 samples, with an average graph depth of 4.4. Each graph yields an average of 7.4 distinct clarification trajectories through traversal, meaning that for each task, we simulate approximately 7 users with different profiles to generate expert clarification dialogues. The complete prompts used for all the above LLM-based steps are shown in Section F.

## A.2. Training Details

*Table 8.* Detailed hyperparameters for different algorithms of IntentRL fine-tuning.

| Training Algorithm | Learning Rate | Batch Size | Number of Training Epochs | Repeat Times | Max Prompt Tokens | Max Response Tokens |
|--------------------|---------------|------------|---------------------------|--------------|-------------------|---------------------|
| GRPO | 1e-6 | 8 | 4 | 8 | 4096 | 1024 |
| SFT | 1e-5 | 64 | 4 | - | 4096 | 1024 |

For reward calculation during training, we utilize `all-mpnet-base-v2` to measure the content score. To compute repetition and deviation penalties, we first use this model to calculate similarities between the generated question and both the interaction history and intent set. If these similarities exceed the repetition threshold $\tau_{rep}$ or fall below the relevance threshold $\tau_{rel}$, respectively, the corresponding penalties are calculated according to Equation 10. Additionally, the format scores and insignificance penalty are calculated through the `Qwen2.5-72B-Instruct` model, and the prompt is shown in Section F.6 and F.7.

As for the hyperparameters for IntentRL fine-tuning, we provide the specific hyperparameters for GRPO and SFT algorithms in Table 8 and all the results of our agent in Section 4 are trained through GRPO algorithm. Furthermore, all experiments were conducted on a cluster of up to 8 NVIDIA H20 GPUs. We utilize the `Trinity-RFT` framework (Pan et al., 2025), a highly customizable RFT training library, to implement our entire workflow, including policy sampling, reward grading, and optimization.

## A.3. Online Simulation

For online evaluation, we propose an intent-aware user simulator to respond to the proactive agent's questions. The user simulator follows a hybrid rule-based and LLM-judge mechanism, as described in Section 3.4. In rule-based logic, we set the repetition threshold $\tau_{rep}$ as 0.92 to judge whether a clarifying question is redundant. Additionally, we set the relevance threshold $\tau_{rel}$ as 0.8 to evaluate the relevance of clarifying question to user real intents. In LLM-judge logic, we prompt `gemini-3-pro-preview` to craft the response in a human-like style. Furthermore, we set the maximum number of clarification rounds to 9, as we demonstrated in Section 4.4 that this configuration yields the highest efficiency and downstream report quality. Finally, we utilize the `Qwen2.5-72B-Instruct` to summarize the information gathered during the clarification process into a refined query for downstream research. The complete prompts of user simulator and summary agent are shown in Section F.8 and Section F.9.

### A.4. Deep Research Agent

For Qwen Deep Research Agent, we use the Dashscope API from Alibaba Cloud to call the model `qwen-deep-research`. It provides a built-in clarification module, which we evaluate as the '+ Qwen clarify' entry in Table 2. For Gemini Deep Research Agent, we utilize `deep-research-pro-preview-12-2025`, the Gemini 3 Pro-powered model that achieves the best performance on Google's recently released DeepSearchQA benchmark. For OpenAI Deep Research Agent, we employ the `o3-deep-research-2025-06-26` version, which we found to exhibit superior performance compared to other available versions. Additionally, we explicitly equip it with two tools provided by OpenAI, including `web_search_preview` and `code_interpreter`. To evaluate its built-in clarification module (the '+ OpenAI clarify' entry in Table 2), we follow the official OpenAI API documentation (OpenAI, 2025b) by deploying a separate `gpt-4.1-2025-04-14` model and applying the prompt template provided in their guidelines. For Tavily Deep Research Agent, we use the `pro` version to generate downstream reports.

## B. More Results

### B.1. Detailed Results on PDR-Bench

*Table 9.* Detailed Scores in P-Score and Q-Score metrics of all baselines on PDR-Bench.

| Method | P-Score | | | | Q-Score | | |
|---|---|---|---|---|---|---|---|
| | GOAL | CONT | PRES | ACTI | DEIN | LOGC | CLAR |
| **Qwen Deep Research Agent** | | | | | | | |
| No clarify | 5.69 | 6.04 | 6.35 | 5.52 | 6.32 | 7.29 | 6.60 |
| + Qwen clarify | 4.71 | 5.09 | 5.03 | 4.22 | 5.04 | 5.71 | 5.14 |
| + Learn-to-Ask clarify | 6.14 | 6.62 | 6.50 | 5.74 | 6.50 | 7.40 | 6.58 |
| + Tell Me More clarify | 6.13 | 6.72 | 6.33 | 5.61 | 6.31 | 7.43 | 6.57 |
| + CollabLLM clarify | 6.23 | 6.71 | 6.56 | 5.71 | 6.58 | 7.51 | 6.75 |
| **+ IntentRL clarify** | **6.28** | **6.76** | **6.61** | **5.77** | **6.64** | **7.62** | **6.77** |
| **Gemini Deep Research Agent** | | | | | | | |
| No clarify | 6.30 | 6.44 | 7.53 | 6.50 | 6.83 | 7.77 | 7.06 |
| + Learn-to-Ask clarify | 6.73 | 7.13 | 7.55 | 6.54 | 7.07 | 7.94 | 7.30 |
| + Tell Me More clarify | 6.77 | 7.34 | 7.49 | 6.58 | 6.83 | 7.87 | 7.12 |
| + CollabLLM clarify | 6.76 | 7.29 | 7.43 | 6.59 | 6.87 | 7.93 | 7.20 |
| **+ IntentRL clarify** | **6.94** | **7.46** | **7.86** | **6.84** | **7.09** | **8.09** | **7.35** |
| **OpenAI Deep Research Agent** | | | | | | | |
| No clarify | 5.39 | 5.53 | 5.59 | 5.06 | 5.85 | 7.05 | 5.88 |
| + OpenAI clarify | 5.91 | 6.37 | 6.26 | **5.86** | 5.87 | 6.82 | **6.55** |
| + Learn-to-Ask clarify | 6.00 | 6.53 | 5.77 | 5.51 | 6.17 | 7.31 | 6.04 |
| + Tell Me More clarify | **6.26** | 6.71 | 5.79 | 5.57 | 6.11 | 7.22 | 5.78 |
| + CollabLLM clarify | 6.17 | 6.64 | **6.50** | 5.65 | 6.21 | **7.35** | 6.03 |
| **+ IntentRL clarify** | 6.23 | **6.84** | 5.96 | 5.74 | **6.28** | 7.33 | 5.98 |
| **Tavily Deep Research Agent** | | | | | | | |
| No clarify | 6.35 | 6.37 | 7.35 | 6.77 | 6.96 | 8.04 | 6.85 |
| + Learn-to-Ask clarify | 6.70 | 7.01 | 7.46 | 6.91 | 7.16 | 8.23 | 7.03 |
| + Tell Me More clarify | 6.79 | 7.14 | 7.28 | 7.09 | 7.13 | 8.18 | 7.20 |
| + CollabLLM clarify | 6.96 | 7.22 | 7.25 | 7.19 | 7.24 | 8.33 | **7.24** |
| **+ IntentRL clarify** | **7.03** | **7.33** | **7.55** | **7.24** | **7.32** | **8.38** | **7.24** |

PDR-Bench provides more detailed sub-dimensions metrics in P-Score and Q-Score to systematically and comprehensively evaluate personalization in DR agents. Specifically, P-Score consists of four sub-dimensions: (1) Goal Alignment (GOAL): evaluates the report's alignment with the user's goals, (2) Content Alignment (CONT): measures the relevance and depth suited to the user's background, (3) Presentation Fit (PRES): assesses whether the report's style and structure matching user preferences, (4) Actionability (ACTI): evaluates the practical value and usefulness of insights. Additionally, Q-Score

includes three sub-dimensions: (1) Depth & Insight (DEIN): measures the analytical richness and originality of the report, (2) Logical Coherence (LOGC): assesses the downstream report's rigor and flow of reasoning, (3) Clarity & Readability (CLAR): evaluates the language fluency and presentation quality of the report. As shown in Table 9, IntentRL *consistently* achieves the highest scores across nearly all sub-dimensions on every DR agent. Notably, the significant improvements in personalization-related metrics compared to the no-clarification baseline confirm that our proactive agent successfully captures user-specific requirements, enabling more tailored report generation. Furthermore, the gains in three sub-dimensions of Q-Score indicate that eliciting deeper user intents also enriches the analytical depth and reasoning quality of reports. While CLAR shows more modest improvements, this aligns with our earlier observation that clarification primarily enhances content relevance rather than surface-level presentation aspects.

### B.2. Results On Original Data from Benchmarks

*Table 10.* Evaluation results of utilizing the original query on DeepResearch Bench and Rigorous Bench.

| Method | DeepResearch Bench | | | | | Rigorous Bench | |
|---|---|---|---|---|---|---|---|
| | Comp. | Insight | Instr. | Read. | Overall | Quality | $1 - $ SDrift |
| w/ IntentRL Clarify | 45.24 | 47.39 | 47.55 | 47.38 | 46.76 | 60.89 | 60.91 |
| w/o Clarify | 43.87 | 45.09 | 46.52 | 46.24 | 45.21 | 59.21 | 59.48 |

In Section 4.1, we mentioned that when evaluating proactive agents on DeepResearch Bench and Rigorous Bench, we employ our query fuzzification and user intent extraction method to simplify and introduce ambiguity into the original queries. However, this preprocessing does not imply that our proactive agent cannot effectively elicit latent user intents from the benchmarks' original, unmodified queries. We conduct an experiment to validate this, as shown in table 10. 'w/o Clarify' denotes the downstream DR agent directly conducting research using the original query, while 'w/ IntentRL Clarify' indicates that our proactive agent first engages in multi-turn clarification to explore missing user intents before research begins. The results demonstrate that our proactive agent *consistently* improves downstream report quality when applied to the original queries from both benchmarks. While these performance gains are relatively modest compared to those observed with simplified queries, this is largely because the original queries in these benchmarks are inherently comprehensive, providing detailed requirements and specific constraints. Therefore, the critical factors determining report quality lie in the search and reasoning capabilities during the research process rather than in supplementing user intent. Nevertheless, the consistent improvements validate that our agent can generalize beyond simplified queries and provide value even when initial user specifications are relatively complete.

## C. Case Study

We present a typical example of our method's training data and clarification process in Figure 6 to demonstrate the evolution of our proactive agent's behavior across the two training stages. In the offline training data, each sample consists of a historical dialogue context paired with a target intent list. During Stage I, we train the agent to generate clarification questions that elicit the intents specified in the target list based on the given context. After Stage I training, our proactive agent demonstrates strong performance in the initial clarification turns. However, as the interaction trajectory progressively deviates from the expert demonstrations in the offline training set, the agent begins to exhibit problematic behaviors in later turns. Specifically, it tends to ask questions that drift off-topic and shows limited sensitivity to user feedback, failing to adjust its strategy to recover from these missteps. This results in repeated questioning patterns, as illustrated in the bottom-left panel where the agent persists with similar questions despite explicit user signals indicating redundancy. After collecting these online rollout trajectories and mixing them into the offline data for Stage II training, the agent's behavior improves substantially in real-world scenarios. As shown in the right panel, the proactive agent becomes more adept at identifying critical missing user intents and formulating high-value questions. Meanwhile, the issue of repetitive questioning is significantly alleviated. Furthermore, when users provide negative feedback (e.g., *"This question is not important to me"*), the agent demonstrates enhanced responsiveness that it promptly adjusts its focus and explores alternative angles that may better align with the user's underlying interests.

We further provide a side-by-side qualitative comparison with other baselines on the same case (*"Discuss the influence ..."*), whose target intent set contains 14 intents in total (denoted as A-N). As shown in Figure 7, Tell Me More terminates clarification immediately without eliciting any target intent and doesn't hit any user intent (Overall: 43.33). Learn to Ask

**Offline Training Data**

**[Context]**
**User:** Develop an evaluation framework for quantitative trading strategies.
**Assistant:** Are you primarily interested in evaluating multi-factor trading strategies ...
**User:** I'm primarily interested in evaluating multi-factor trading strategies, but I want to make sure ...

**[Target Intent]**
1. ... enable accurate comparison and analysis of strategies across…
2. ... concerned that the market currently lacks a single …
3. ... comprehensively define the 'returns' dimension for quant strategies …

**Clarification Process after Stage 1 Training**

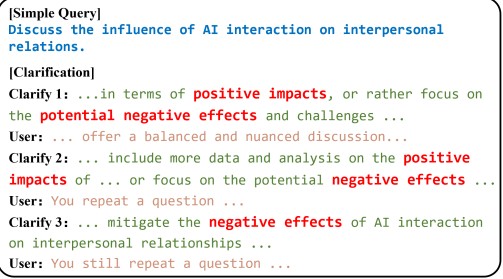

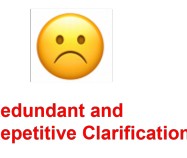

**Redundant and Repetitive Clarification**

**Clarification Process after Stage 2 Training**

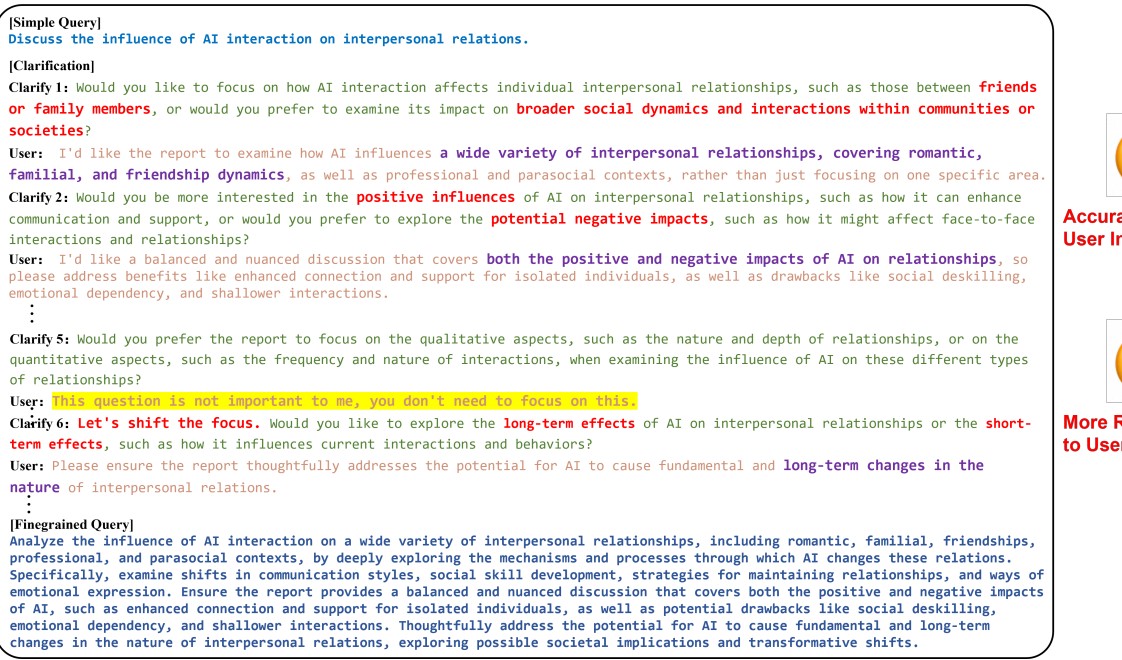

**Accurately Capturing User Intent**

**More Responsiveness to User Feedback**

*Figure 6.* Case study of the clarification details of our proactive agent after 2 stages training.

identifies an initial relevant intent, but after receiving negative feedback on an irrelevant query, it rephrases essentially the same question, indicating limited sensitivity to user feedback. This trajectory covers 3 intents and obtains 45.76 downstream overall scores. CollabLLM explores the intent space more effectively and reaches 5 intents and 46.95 downstream overall scores, but it still exhibits weaker implicit intent exploration ability than our method. In contrast, our IntentRL captures 7 intents, achieving the best overall score. This comparison suggests that capturing a denser set of intents during clarification directly enriches the downstream report, driving higher rubric hits and overall scores.

## D. Specific Example of C-DAG Structure

A specific example of a constructed C-DAG from our training data is shown below in JSON format. Specifically, the `root_node` represents the simplified query, while `start_node_ids` specifies the depth-1 nodes in the C-DAG, which serve as the initial nodes in the traversal stack. The `nodes` field stores the specific content and options for each clarification question. Each option contains a `next_node_id` field that points to its child node, meaning that when the current node is visited and the user selects this option, the corresponding child node is added to the traversal stack as an accessible node for subsequent clarification turns.

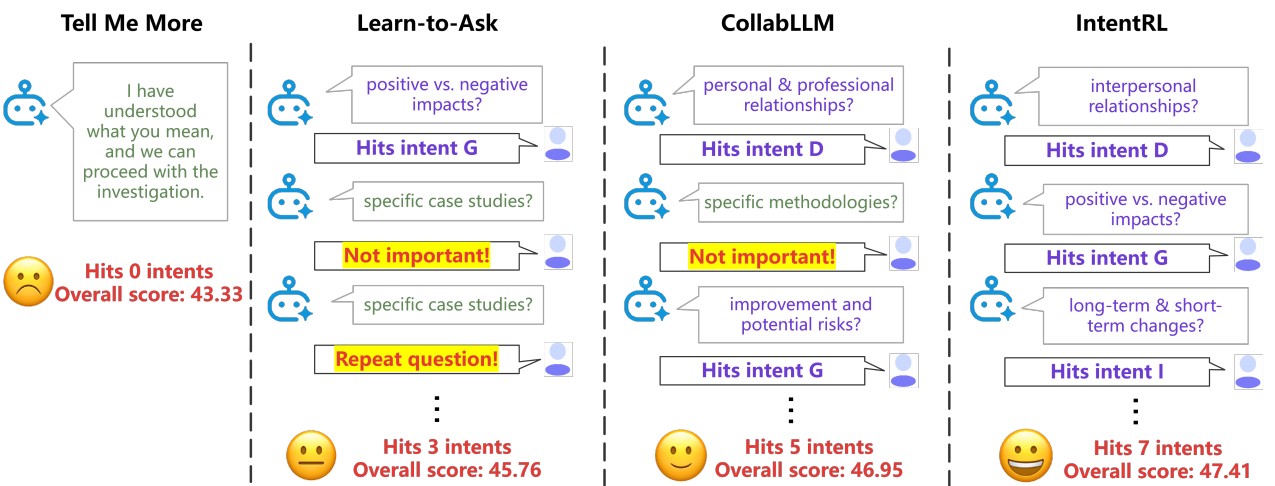

*Figure 7.* Case study of clarification trajectories across different baselines.

Example of C-DAG Structure in JSON Format colback

```
{
  "root_node": "Summarize the involvement of major international consulting firms in
  ↪  Artificial Intelligence (AI).",
  "start_node_ids": [
    "q1",
    "q2"
  ],
  "nodes": {
    "q1": {
      "id": "q1",
      "text": "Which specific group of consulting firms would you like to focus on?",
      "options": [
        {
          "text": "The Big Four, Accenture, and MBB",
          "next_node_id": []
        },
        {
          "text": "IBM and Capgemini",
          "next_node_id": []
        },
        {
          "text": "A representative range including all of the above and similar
          ↪  firms",
          "next_node_id": []
        }
      ]
    },
    "q2": {
      "id": "q2",
      "text": "What is the primary focus of your research on these firms' AI
      ↪  involvement?",
      "options": [
        {
          "text": "Their external activities and market offerings",
          "next_node_id": [
            "q3",
            "q11"
          ]
        },
        {
          "text": "Their internal strategy and development",
          "next_node_id": [
            "q4",
            "q11"
          ]
```

```
          }
        ]
      },
      "q3": {
        "id": "q3",
        "text": "Regarding their external activities, what information are you more
    ↪  interested in?",
        "options": [
          {
            "text": "Global investments, key initiatives, and outputs",
            "next_node_id": [
              "q5",
              "q12"
            ]
          },
          {
            "text": "AI-driven products, services, and client case studies",
            "next_node_id": [
              "q6",
              "q7",
              "q12"
            ]
          }
        ]
      },
      "q4": {
        "id": "q4",
        "text": "Regarding their internal strategy, which aspect is more important to
    ↪  you?",
        "options": [
          {
            "text": "Their strategic directions in AI",
            "next_node_id": [
              "q8",
              "q12"
            ]
          },
          {
            "text": "Their talent development programs for AI",
            "next_node_id": [
              "q9",
              "q12"
            ]
          }
        ]
      },
      "q5": {
        "id": "q5",
        "text": "When analyzing AI investments, what is more important to understand?",
        "options": [
          {
            "text": "The scale, nature, and global scope of financial commitments",
            "next_node_id": []
          },
          {
            "text": "The strategic rationale and meaningful patterns behind these
    ↪  investments",
            "next_node_id": []
          }
        ]
      },
      "q6": {
        "id": "q6",
        "text": "When covering AI-driven offerings, what is the desired level of
    ↪  analysis?",
        "options": [
          {
            "text": "A comprehensive overview of their products, platforms, and
    ↪  services",
            "next_node_id": []
          },
          {
```

```
            "text": "A critical assessment of the value, innovativeness, and client
            ↪   impact of these offerings",
            "next_node_id": []
          }
        ]
      },
      "q7": {
        "id": "q7",
        "text": "How should the report approach client case studies and application
        ↪   scenarios?",
        "options": [
          {
            "text": "Provide a diverse range of examples across different industries",
            "next_node_id": []
          },
          {
            "text": "Extract deep insights on practical impact, scalability, and key
            ↪   success factors from key cases",
            "next_node_id": []
          }
        ]
      },
      "q8": {
        "id": "q8",
        "text": "Regarding their AI strategy, what is the preferred analytical
        ↪   approach?",
        "options": [
          {
            "text": "Detail each firm's individual vision and main focus areas",
            "next_node_id": []
          },
          {
            "text": "Identify overarching strategic themes and compare their competitive
            ↪   positioning",
            "next_node_id": []
          }
        ]
      },
      "q9": {
        "id": "q9",
        "text": "When discussing AI talent, which aspect is more critical to cover?",
        "options": [
          {
            "text": "How they acquire and retain external AI talent",
            "next_node_id": []
          },
          {
            "text": "Their initiatives to train and upskill their existing workforce",
            "next_node_id": []
          }
        ]
      },
      "q11": {
        "id": "q11",
        "text": "What should be the core analytical lens of the report?",
        "options": [
          {
            "text": "Analyze each firm's AI components (strategy, talent, products)
            ↪   separately",
            "next_node_id": []
          },
          {
            "text": "Focus on how these components are aligned and mutually reinforce
            ↪   each other",
            "next_node_id": []
          }
        ]
      },
      "q12": {
        "id": "q12",
        "text": "What should be the concluding focus of the report?",
        "options": [
```

```
            {
                "text": "A summary of the current state of AI involvement across the firms",
                "next_node_id": []
            },
            {
                "text": "Identifying emerging trends, future directions, and significant
                ↪  implications",
                "next_node_id": []
            }
        ]
    }
  }
}
```

# E. Limitations

While IntentRL demonstrates strong performance across multiple benchmarks, we acknowledge limitations regarding its generalizability to unconstrained, real-world user interactions. Our training pipeline relies on a synthetic clarification graph (C-DAG) and an online user simulator grounded in fixed, explicit target intents. While this design provides consistent feedback for agents to adapt and mitigate exposure bias in out-of-distribution scenarios, it does not fully capture the ambiguity, inconsistency, or preference shifts commonly observed in real users. As a result, the learned policy may be less reliable in settings where user intents are only partially expressible, fuzzy, or internally contradictory. However, this limitation is not unique to our method, but is common in current simulation-based RL for proactive or user-centric agents (e.g., UserRL (Qian et al., 2025b)). In such settings, controllable intent representations and simulator consistency are often necessary to make policy optimization tractable, but they inevitably simplify the sim-to-real interaction gap.

Therefore, we view closing this gap as an important direction for future work. One natural next step is to combine simulator-based training with periodic human-in-the-loop feedback, allowing the policy to adapt to real user behavior after obtaining a stable initialization in simulation. Another useful extension is to make the simulator less deterministic by explicitly modeling vague or conflicting preferences, for example through stochastic intent sampling or contradictory intent sets, which may better reflect the uncertainty and inconsistency of real users. A more direct way to study sim-to-real transfer is to conduct controlled user studies in which participants interact with the agent on open-ended research tasks under a standardized protocol, while measuring not only final report quality but also interaction efficiency, the usefulness of clarification, and user satisfaction throughout the dialogue. Such studies could help identify which aspects of real user behavior are missing from the current simulator and provide clearer guidance for improving robustness at deployment time.

# F. Used Prompts

## F.1. Query Simplification & Shallow Intent Extraction

> **Prompt for Query Simplification & Shallow Intent Extraction**
>
> You are a query analysis and transformation expert. Your task is to receive an "original query" and perform the following operations:
> 1. Simplify Query: Convert the original query into a more concise and broader 'simple_query'. This 'simple_query' should only retain the most core, high-level topic of the original query.
> 2. Extract Intent: Convert all the information that was simplified or removed into a 'missing_intent' list. This list should be expressed from the user's first-person perspective (e.g., using "I want to..." or "I need to...").
>
> # Original Query
> {task}
>
> # Important Rules
> 1. Maintain a Valid Query Format: The 'simple_query' must remain a well-formed user request. It should preserve the original's core action (e.g., "analyze," "collect," "research") or question structure (e.g., "what is," "how does"). Do not reduce it to a mere topic phrase or a few keywords. For example, simplify "Collect data on X and analyze Y"

to "Research the relationship between X and Y," not just "X and Y relationship."

2. If the original query is already very short, specific, and has no extra details that can be simplified (e.g., it is a single, direct technical question), then return an empty string for 'simple_query' and an empty list for 'missing_intent'.

3. Your output language must be the same as the original query's language. For instance, if the original query is in English, your output for the simple query and missing intents must also be in English. Conversely, if the original query is in Chinese, your output must also be in Chinese.

# Examples
1.
Original Query: Collect and compile the actual income and financial status of China's 9 social strata, with particular focus on researching and identifying the characteristics of China's middle class, including the actual number of middle-class individuals, their financial capacity, etc.

Simple Query: Collect and compile the economic situation of China's 9 social strata

Missing Intent: [I want to collect the actual income and financial status of China's 9 social strata, I want to particularly research the characteristics of China's middle class, including the actual number of middle-class individuals, their financial capacity, etc.]

2.
Original Query: Research the relationship between investment and lending relationships among domestic financial institutions and systemic risk? Model the lending relationships and risks at different levels or types

Simple Query: Research the correlations among domestic financial institutions

Missing Intent: [I want to research correlations including the connection between investment and lending relationships and systemic risk, I want to model the lending relationships and risks at different levels or types]

3.
Original Query: Collect and compile relevant information on the current top ten insurance companies by international comprehensive strength, conduct a horizontal comparison of various dimensions including each company's financing situation, credibility, growth rate over the past five years, actual dividends, future development potential in China, etc., and evaluate for me the 2-3 companies most likely to rank high in assets in the future

Simple Query: Collect and compile relevant information on the current top ten insurance companies by international comprehensive strength

Missing Intent: [I want to conduct a horizontal comparison of various dimensions including each company's financing situation, credibility, growth rate over the past five years, actual dividends, future development potential in China, etc., I want an evaluation of the 2-3 companies most likely to rank high in assets in the future]

# Output Format: You must strictly adhere to the following JSON format for your response.
{
"simple_query": "SIMPLE_QUERY_STRING",
"missing_intent": [
"MISSING_INTENT_1",
"MISSING_INTENT_2"
]
}

## F.2. Rubric-derived Deep Intent Generation

---

**Prompt for Rubric-derived Deep Intent Generation**

You are an intelligent assistant specializing in converting evaluation rubrics into a clear, direct list of user intents.
# Task
Given a JSON-formatted rubric below that describes the elements of a high-quality report. You will convert these evaluation criteria into a list of intents from a first-person user's perspective.

# Rubrics
{rubrics_str}

# Important Rules
1. Combine each criterion and its explanation to distill a specific, clear user intent. Ensure that all key information points from the original rubric are included.
2. Use the first-person perspective, emulating a user who knows what they want. For example, use phrases like "I want the report to...", "The report needs to clarify...", and "Please ensure...".
3. Your output language must match the language of the original query. That is, if the original query is in English, your output intents must also be in English. Conversely, if the original query is in Chinese, your output intents must also be in Chinese.

# Output Format: Please strictly adhere to the following JSON format for the result. List all criteria from the rubric in their original order.
{
"comprehensiveness": [INTENT_1, INTENT_2, ...],
"insight": [INTENT_1, INTENT_2, ...],
}

---

## F.3. Base Graph Construction

---

**Prompt for Base Graph Construction**

# Background
The goal of the Deep Research Agent is to generate a comprehensive, in-depth, high-quality research report based on a single task request from the user. To ensure the correct direction before research begins, the agent has the capability to conduct multiple rounds of follow-up questions with the user before executing the task, uncovering potential intents that the user has not explicitly expressed in the task request. These follow-up questions are organized into a "Clarification Directed Acyclic Graph" structure, which gradually refines and clarifies task requirements through interaction with the user.

# Task
Your core task is to design the initial version of this clarification Directed Acyclic Graph based on the given original coarse-grained task and the list of user's potential needs. Specific steps are as follows:
1. Design Questions and Options: Iterate through the user's potential needs list, converting each item into a clear and specific clarification question directed at the user, along with 2-3 selectable options. Options can be extracted from text such as "for example," "including," "X and Y," "X or Y" within that item. For example, the intent item "want to collect the actual income and financial status of China's 9 social strata" can be extracted into the question "Which aspect of the economic situation of China's 9 social strata are you more concerned about?" with two options: "actual income" and "financial status."
- Note: All option content must originate from the user's potential needs, which means each option represents a valid user requirement. You cannot fabricate content beyond the user's potential needs as options.
2. Build Clarification Directed Acyclic Graph Structure: Organize all questions and options into a clarification Directed Acyclic Graph in JSON format. You need to define:

---

* Which questions are parallel initial questions (starting nodes).
* Which questions have dependency relationships (one question's answer determines the next question).
* Ensure the final output JSON structure strictly follows the format in "Output Requirements" below.

# Input Data
1. Original Coarse-grained Task
{task}
2. User's Potential Needs List
{intent_list_str}

# Output Requirements
You can only generate one JSON object, without any other additional content. This object represents the clarification Directed Acyclic Graph. Its structure must follow the example format below (you do not need to write comments). 'start_node_ids' defines the starting question nodes with no prerequisites. The 'nodes' object contains definitions of all question nodes, and the 'next_node_id' list defines the subsequent question nodes that will be triggered after selecting an option.
'Example Directed Acyclic Graph'

# Important Rules
- Your output language must be the same as the language of the original coarse-grained task. If the original task is written in Chinese, then the generated follow-up questions and option content should also be in Chinese. Similarly, if the original task and scoring criteria are written in English, then the generated follow-up questions and option content should also be in English.
- The number of options corresponding to each question must strictly be no fewer than 2 and no more than 3!

## F.4. Graph Expansion

**Prompt for Graph Expansion**

# Task
You have already constructed an initial clarification Directed Acyclic Graph. Now, you need to expand the existing clarification Directed Acyclic Graph by adding deeper-level follow-up questions. Your core task is to deepen and expand the existing clarification Directed Acyclic Graph based on a list of deeper, more fine-grained user intents, increasing its branches and depth, thereby building a more complex clarification path that can comprehensively capture user intents, while ensuring that there is no content duplication between nodes.

# Input Data
1. Initial Clarification Directed Acyclic Graph: A simple clarification Directed Acyclic Graph with only a few nodes or even no nodes
{original_Directed Acyclic Graph}
2. User Intent List: Contains all complete requirements of the user for this task, some of which may require online search to complete
{intent_list_str}

# Step-by-Step Guide
1. Carefully read and review each item in the user intent list, determining which items are more suitable for obtaining information by asking the user, and which items are more suitable for obtaining information through web search. Avoid asking the user about knowledge that needs to be researched and searched, which would reduce the user experience.
2. Filter out at least 8 intents that are most suitable for obtaining information by asking the user. These often can influence the research direction and establish the foundation for research scope, standards, and focus.
3. Iterate through all intents filtered in step 2, converting each item into a clear and specific clarification question

directed at the user, along with 2-3 selectable options. Options can be extracted from text such as "for example," "including," "X and Y," "X or Y" within the intent. For example, the intent item "want to collect the actual income and financial status of China's 9 social strata" can be extracted into the question "Which aspect of the economic situation of China's 9 social strata are you more concerned about?" with two options: "actual income" and "financial status."

- Note: All option content must originate from the user's potential needs, which means each option represents a valid user requirement. You cannot fabricate content beyond the user's potential needs as options.

4. If the initial clarification Directed Acyclic Graph does not have any nodes, then you directly construct a new clarification Directed Acyclic Graph; if the initial clarification Directed Acyclic Graph has nodes, then you need to expand the clarification Directed Acyclic Graph. The node addition steps are as follows:

- Understand the current Directed Acyclic Graph's design and each follow-up question within it
- Iterate through each node, trying to design different follow-up questions for each of its options. These follow-up questions should also play an important role in clarifying user intent and improving the quality of the final report
- Design 2-3 options for this follow-up question node
- Design different follow-up questions for each option
- Continuously and recursively repeat steps 2-4 until a follow-up question subDirected Acyclic Graph that can clearly clarify user intent is constructed for each newly added node. When designing a subDirected Acyclic Graph for a node, you can dig deeper based on a particular intent, which is equivalent to designing an increasingly in-depth path for the research, thereby improving the depth of the overall research process.
- You only need to generate one expanded clarification Directed Acyclic Graph JSON object at the end, without any other additional content.

Important Rules:

1. Carefully check the final generated Directed Acyclic Graph, and delete any nodes with highly overlapping content.
2. The number of options corresponding to each question must strictly be no fewer than 2 and no more than 3! Try to have different options correspond to different follow-up questions to ensure the diversity and density of Directed Acyclic Graph nodes.
3. The total number of nodes in the expanded Directed Acyclic Graph cannot exceed 20.
3. Your output language must be the same as the language of the user intent list. If the user intent list is written in Chinese, then the generated follow-up questions and option content should also be in Chinese. Similarly, if the user intent list is written in English, then the generated follow-up questions and option content should also be in English.

The method of adding nodes can refer to the following example. You do not need to write comments when generating the Directed Acyclic Graph.
Original Directed Acyclic Graph:

{{
"start_node_ids": ["q1", "q2"],
...
}}

Expanded clarification Directed Acyclic Graph JSON object. It should contain all nodes from the initial Directed Acyclic Graph and add new nodes and connection relationships to reflect the exploration of deeper user intents. The format should be consistent with the example below.

{{
"start_node_ids": ["q1", "q2", "q3"],
...
}}

## F.5. System Prompt for Proactive Agent

---

**System Prompt for Proactive Agent**

# Task
You are an AI assistant specializing in user intent clarification. Given an ambiguous user request, your primary goal is to identify the **single most critical** piece of missing information and ask a targeted question to resolve it.

# Guidelines
1. Your response must be a **single paragraph** containing exactly one concise question with 2-3 distinct answer choices. Note that don't provide too many choices, which will make users feel complicated.
2. **NEVER** repeat a question that has already been asked in the conversation history. If a user points out that your question is repetitive, you must immediately pivot to a new line of questioning.
3. Your question must not be a simple rephrasing of the user's request or breaking it down into the components they already mentioned. Instead, it should seek to uncover a crucial piece of underlying context, such as the user's goal or a specific constraint they haven't stated.
4. If the user indicates your question is **NOT Relevant or Important**, re-analyze their request from a different angle and ask a new, valuable question to uncover a different critical piece of information.
5. Respond in the same language as the user's request. For instance, if the user asks in English, you must reply in English; if the user ask in Chinese, you must reply in Chinese.

---

## F.6. Prompt for Format Scoring Reward Calculation

---

**Prompt for Format Scoring Reward Model**

You are tasked with evaluating the format of a question by counting how many distinct questions it contains and assigning a format score accordingly.

# Input
{}

# Scoring Rubric
- 1.0: The input contains exactly 1 question with multiple options/choices
- 0.5: The input contains exactly 2 questions with options/choices
- 0.0: The input contains more than 2 questions

# How to Count Questions
**Key principle**: Count the number of question marks (?) in the input. Each question mark typically indicates one distinct question.

**Important distinctions**:
- A question may present multiple options or choices (e.g., "option A or option B"). These options are NOT separate questions; they are alternative answers to the same question.
- Options are typically connected by words like "or", "and", or presented as alternatives within a single interrogative sentence.
- Only count distinct questions, not the number of choices/options provided.

- Question marks: 2
- Reasoning: This contains 2 distinct questions (one about focus areas, another about opportunities vs challenges)
- Score: 0.5

# Output Format
<think>Explain your reasoning for the format and content scores clearly and concisely. </think>

---

<format_score>Insert only the format score as a float (e.g., 1.0, 0.5, 0.0)</format_score>

> Important:
> - Output **exactly** the two tags shown above.
> - Do **not** include any additional text, explanation, or formatting outside the tags.
> - Ensure clarity and precision in your evaluation reasoning within the '<think>' tag.

## F.7. Prompt for Insignificance Penalty Calculation

---

**Prompt for Insignificance Penalty Model**

You are a Quality Assurance specialist for AI-human interaction. Your task is to evaluate whether an AI's follow-up question is "valid" or "redundant".
# Task
Analyze the provided User Request and the AI's Clarification Question to determine if the AI is genuinely seeking new context or simply "parroting" the user.

# Evaluation Criteria
A follow-up question is **REDUNDANT** if:
- Self-Answering: It asks the user to provide the answer to their own original question (e.g., User asks "Which is better, A or B?", AI asks "Do you think A or B is better?").
- Restating Requirements: It asks the user to choose between components they have already explicitly requested to be included (e.g., User asks for "A and B," AI asks "Do you want A or B?").
- Zero Information Gain: It rephrases the request as a question without seeking any underlying constraints, goals, or missing parameters.

# Examples
**Example 1:**
- User Request: "Collect and compile the actual income and financial status of China's current 9 social strata, with particular focus on researching and identifying the characteristics of China's middle class, the actual number of middle-class individuals, their financial capacity, etc."
- Clarification Question: "When researching the income and financial status of China's middle class, which aspect are you most concerned about? A. Income level B. Financial status"
output:
<think>The Clarification Question asks the user to choose between aspects (income and financial status) they already explicitly mentioned wanting to research.
</think>
<verdict>1</verdict>

**Example 2:**
- User's Request: "Future development trends of China's finance sector, which subdivided field (such as investment banking, PE, fixed income, etc.) has more upward potential in the future"
- AI Follow-up: "Among the future development trends in China's finance sector, which subdivided field (such as investment banking, private equity (PE), fixed income, etc.) do you think has greater upward potential? A. Investment banking... B. Private equity... C. Fixed income..."

# Output Format:
<think>A concise explanation of why the question fails or succeeds</think> <verdict>Insert 1 (redundant) or 0 (valid)</verdict>

---

## F.8. User Simulator

> **Prompt for User Simulator**
>
> You are role-playing as a human USER interacting with an AI collaborator to complete a specific task. Your goal is to generate realistic, natural responses that a user might give in this scenario.
>
> # Input Information:
> You will be provided with an Intent List
> {}
>
> # Task Description:
> Given the ongoing conversation between you (as the user) and the AI assistant, your task is to answer the question in the last message from assistant into a natural, conversational response that a human user would provide.
> 1. Analyze the question to determine which intent from the Intent List it corresponds to. This intent reflects the underlying goal behind the answer.
> 2. Use the chosen intent as a guide to craft an answer in a human-like style. Your response should sound like something a person would actually say, rather than a robotic selection of an option or a direct statement of fact.
>
> ## Guidelines:
> - Stay in Character: Role-play as a human USER. You are NOT an AI. Maintain a consistent personality throughout the chat.
> - Goal-Oriented: Keep the chat focused on your intent. Avoid small talk or digressions. Redirect the chat back to the main objective (your Intent List) if it starts to stray.
> - Don't Copy Input Directly: Use the provided information for understanding context only. Avoid copying target queries or any provided information directly in your responses.
> - While your response should be creative and not a direct copy, it must incorporate every detail information from the chosen intent.
>
> ## Output Format
> {Your response answer}
>
> > Important:
> > - Only output your response answer. Do **not** include any additional text, explanation, or formatting in your response.
> > - Phrase your response as a declarative statement, not a question.
> > - Your output language must match the language of your chat history. That is, if the chat history is in English, your output rephrased answer must also be in English. Conversely, if the chat history is in Chinese, your output rephrased answer must also be in Chinese.

## F.9. System Prompt for Summary Agent

> **System Prompt for Summary Agent**
>
> You are an expert Query Optimizer. Your task is to transform a simple, vague original query into a comprehensive, high-quality finegrained query based on your history clarification dialogue.
>
> # Input Information
> 1. Original Query: {original_query}
> 2. History Clarification Dialogue:
> {history_clarification}
>
> # Guidelines

1. Carefully review your history clarification. Identify every constraint, preference, and detail in the conversation.
2. Merge the core topic of the original query with the specific details extracted from the dialogue.
3. Guarantee absolute information fidelity. The refined query must be strictly and exclusively derived from the original query and the valid clarification dialogue. Do not introduce any external information, assumptions, conceptual substitutions, fabricated details, or excessive stylistic embellishments.
4. Rewrite the prompt into a single, cohesive, and professional query.

# Information Notes
- Perfectly styled as a direct, imperative user command (e.g., starts with "Analyze...", "Create...", "Act as..."). It is written from the user's perspective, is fully actionable, and contains no conversational AI-like phrasing.
- The language should be precise and adapted to the target audience mentioned in the dialogue (if any).
- Ensure there are no logical conflicts.
- Exclude information from any clarification exchanges the user has identified as irrelevant, unimportant, or repetitive. Do not incorporate such dismissed information into the final query.
- Your output language must match the language of the Original Query. That is, if the Original Query is in English, your output 'finegrained_query' must also be in English. Conversely, if the Original Query is in Chinese, your output 'finegrained_query' must also be in Chinese.

## Output Format
Your response must be a **single paragraph** containing **ONLY** the text of the 'finegrained_query'. Do not include explanations.

