# OpenReview forum: "IntentRL: Training Proactive User-intent Agents for Open-ended Deep Research via Reinforcement Learning"
_ICML.cc/2026/Conference — ICML 2026 regular_

### Official Review · Reviewer_7jpW · 2026-03-09

**Soundness:** 3
**Presentation:** 3
**Significance:** 2
**Originality:** 2
**Overall Recommendation:** 4
**Confidence:** 3

**Summary:**

This paper focuses on Deep Research (DR) agents addressing the “autonomy-interaction dilemma.” The authors propose Clarification DAG (C-DAG) for scalable data construction, and IntentRL, a two-stage RL training framework that trains proactive agents to clarify latent user intent before executing long-horizon research. The experiment and analysis results demonstrate the proposed framework’s effectiveness.

**Compliance With Llm Reviewing Policy:**

Affirmed.

**Final Justification:**

Many of the concerns raised appear to stem from ambiguities or incomplete explanations in the paper that could lead to misunderstandings of the authors’ intentions. I believe these issues can be sufficiently addressed during the camera-ready revision.

**Key Questions For Authors:**

Q1. Reward validity.
The content reward in Equation 7 measures cosine similarity between the generated question and the target intent embeddings, rather than between the user's answer and the intent. This means the agent is rewarded for asking semantically aligned questions, regardless of whether those questions actually elicit the intended information. It would be beneficial for the authors to add justification on whether question-intent alignment ensures that the question actually elicits the intended information from the user.

Q2. Summary agent's contribution to downstream performance.
The authors used Qwen2.5-72B-Instruct as their summary agent, but there is no ablation isolating this component's contribution. I'm curious about the effect of the summary agent's capability. It would be great if the authors could provide an ablation comparing summary agents with smaller models or directly passing raw dialogue history.



Q3. User simulator fidelity.
The online simulator uses fixed ground-truth latent intents, making it more of an oracle than a realistic user, where real-world users might have fuzzy and sometimes contradictory preferences. It would be interesting to hear the authors' thoughts on how this gap between the simulated and real user behavior might affect the framework's performance in practice.

**Limitations:**

yes

**Strengths And Weaknesses:**

Strength

S1. Well-motivated problem.
DR agents have recently gained significant attention, and this paper tackles an interesting phenomenon, the "autonomy-interaction dilemma", which is very interesting. The Introduction convincingly establishes the background and significance of this problem, making a compelling case for the proposed approach.

S2. Strong empirical results.
The experimental results effectively demonstrate the proposed framework's effectiveness. In particular, the reasonable reimplementation of proprietary systems such as Qwen clarify and OpenAI clarify as baselines strengthens the validity of the comparisons and further supports the effectiveness of the proposed approach.


S3. Practical plug-and-play design.
The framework is designed to be model-agnostic and can be seamlessly integrated with a wide range of downstream DR agents. This plug-and-play property is expected to have broad practical utility and meaningful impact on the field.


Weakness


W1. The POMDP formulation requires further justification.
The authors formulate intent clarification as a POMDP where the user's latent intent $I$ is the hidden state. However, the paper explicitly states that $I$ is "fixed throughout the dialogue," which can be translated as there is no state transition dynamic, which is the defining challenge that POMDP is designed to address. It remains unclear how the proposed formulation and training procedure differ from standard MDP. It will be great for the authors to clarify this point.

W2. The claimed scalability is bounded by the seed dataset size.
The paper highlights scaling 50 seed samples into 2,347 dialogue turns as a key contribution. However, this expansion operates only within each task, generating diverse clarification trajectories over the same fixed set of topics. Since C-DAG construction requires both an original query and its corresponding rubric, extending the framework to new tasks still demands human-curated data, which limits the claimed scalability. It would be helpful if the authors could address how the proposed pipeline extends to new task domains, and whether the dependency on human-curated rubrics can be reduced.

W3. The training algorithm selection lacks stage-specific justification.
Figure 4 shows that SFT outperforms GRPO in offline testing while GRPO dominates in online evaluation. Given that Stage 1 targets offline expert imitation and Stage 2 targets online exploration, the most natural design would be to apply SFT for Stage 1 and GRPO for Stage 2. However, this hybrid combination is never explored, leaving the choice of GRPO for both stages unjustified. Furthermore, the analysis in Section 4.4 compares SFT and GRPO by applying each uniformly across both stages, which conflates two fundamentally different learning objectives. It would be helpful if the authors could provide a stage-specific algorithm analysis and clarify the rationales behind the design choices.

---

> ### Author Rebuttal · Authors · 2026-03-31
>
> Thanks for your valuable suggestions! We provide the following responses point by point:
> > W1: POMDP formulation
>
> There may be a misconception that POMDPs are defined by **partial observability rather than dynamic hidden states**. POMDPs with static hidden states are standard for active information-gathering tasks like HiP-MDPs. In our formulation, the fixed latent intent $I$ remains unobservable to the agent, necessitating incremental inference through interactions. This exemplifies the typical partial observability challenge POMDPs address. Furthermore, the formulation is **not transition-free** because the observable history $H_t$ evolves per turn, and user responses follow stochastic transitions $u_t \sim P(\cdot|I, H_{t-1}, x_t)$. The inability to directly observe $I$ fundamentally distinguishes this from standard MDPs. We will elaborate on this in Section 3.1.
> > W2: Framework scalability
>
> Our scalability claim emphasizes that reduced annotation costs make our pipeline adaptable to other data-scarce tasks.  Regarding generalization, IntentRL trained solely on DR Bench achieves strong performance on Rigorous Bench and PDR-Bench. Both domains differ significantly from DR Bench and require no additional rubrics or task-specific annotations beyond evaluation needs.
>
> Additionally, while rubrics are required as seeds, we consider this practical since 1) many DR benchmarks provide structured rubrics, 2) LLMs can generate rubrics with minimal oversight, and 3) our cross-domain results demonstrate learned clarification strategies transfer to unseen tasks without rerunning the data construction pipeline.
> > W3: Training algorithm
>
> IntentRL diverges from the trending SFT cold-start → RL paradigm and Stage II is not initialized from the Stage I checkpoint. **Our objective is goal-oriented exploration grounded in C-DAG traversal states rather than trajectory fitting**. Instead of imitating expert trajectories for behavior cloning, Stage I employs a hindsight-driven strategy using offline data to train the agent to explore remaining intents to be covered from the current state. Consequently, GRPO is more appropriate than SFT in Stage I due to its exploration incentives. To address offline training limitations, we introduce semi-online rollouts in Stage II. We conducted an additional SFT+GRPO experiment on DR Bench with Qwen DR agent. Results indicate it underperforms our pipeline in online evaluation (downstream report quality), confirming GRPO is essential in Stage I.
> Strategy|Comp.|Insight|Instr.|Read.|Overall
> -|-|-|-|-|-
> SFT+GRPO|41.34|43.46|42.12|47.17|43.06
> Ours|41.49|44.06|43.19|47.04|43.65
>
> > Q1: Reward validity
>
> There may be a misunderstanding of how simulator operates. The simulator needs to first calculate question-intent similarity to verify semantic alignment with an intent via a relevance threshold, then generate an intent-grounded answer only if this threshold is met. Thus, question-intent alignment acts as the gating condition triggering user responses.
>
> Furthermore, rewarding question-intent similarity directly ensures **preciser credit assignment** since answer rewarding entangles with LLM linguistic randomness. Rewarding agent direct actions maintains pure and immediate reward signals isolated from environmental variance. Actually, cooperative users naturally provide intended information when asked expected questions.
> > Q2: Summary agent
>
> We conduct ablation on DR Bench comparing Qwen2.5-72B, Qwen2.5-7B and directly passing raw dialogue history to the downstream DR agent.
> Model|Comp.|Insight|Instr.|Read.|Overall
> -|-|-|-|-|-
> Qwen2.5-72B|41.49|44.06|43.19|47.04|43.65
> Qwen2.5-7B|41.37|44.06|43.12|46.37|43.53
> no summary|41.73|43.96|43.24|45.79|43.51
>
> Results indicate summary agent capacity has minimal impact. Qwen2.5-7B achieves 43.53 versus 43.65 for 72B, and passing raw dialogue history yields 43.51. This demonstrates IntentRL gains stem primarily from high-quality intent clarification rather than summary agent capability. Nonetheless, we recommend stronger summary models when feasible to produce precise refined queries for downstream DR agents.
> > Q3: User simulator fidelity
>
> We acknowledge the simulator gap with real users. However, the primary motivation of our simulator is to provide consistent feedback, **teaching agents to adapt and mitigate exposure bias in out-of-distribution scenarios rather than perfectly modeling human fuzziness**. To enhance realism, we prompt responses in a human-like style with consistent personalities. Furthermore, PDR-Bench, which features authentic, complex user profiles (like interests, habits...) instead of synthetic rubric-derived intents, shows significant gains, especially on personalization dimensions (GOAL, CONT). This proves our strategy transfers well to real-world diversity.
>
> In fact, this challenge remains a shared limitation in simulation-based RL (e.g., UserRL, Qian et al.). We will note closing this gap as a future direction in the limitations section.

---

> > ### Author Rebuttal · Reviewer_7jpW · 2026-04-03
> >
> > Thank you for the detailed and thoughtful responses. I will raise my score accordingly.

---

### Official Review · Reviewer_KMHY · 2026-03-12

**Soundness:** 3
**Presentation:** 3
**Significance:** 3
**Originality:** 3
**Overall Recommendation:** 4
**Confidence:** 4

**Summary:**

This paper addresses an often overlooked issue in multi-turn deep search agents: the misunderstanding of user intent. To mitigate this, the authors introduce IntentRL, a framework designed to scale user interactions instead of search space, thereby efficiently increasing the overlap between user intent and the search results. To accomplish this, the paper introduces a two-stage RL framework to train proactive agents to ask targeted clarification questions. To obtain training data, the authors define both shallow and deep intent and construct a Clarification DAG to systematically generate interaction trajectories. Experiments on both in-domain and out-of-domain benchmarks demonstrate the performance improvement of IntentRL compared to established baselines.

**Compliance With Llm Reviewing Policy:**

Affirmed.

**Final Justification:**

The rebuttal adequately addresses my concerns.

**Key Questions For Authors:**

- Q1: How do you train proactive agents with SFT using the step 2 online strategy?
- Q2: Could you share the training curves for SFT? Since the results suggest overfitting, what is the SFT performance after 1, 2, and 3 epochs respectively?

**Limitations:**

yes

**Strengths And Weaknesses:**

**Strengths:**

- S1: The research question is highly relevant and timely, particularly for deep search agents that undergo extensive reasoning paths where intent drift can occur.
- S2: The introduction of the Clarification DAG is a novel and structured way to bridge the gap between static datasets and the dynamic nature of multi-turn clarification.
- S3: The paper provides a clear distinction between shallow and deep intent, offering a more nuanced framework for evaluating agent-user communication.

**Weaknesses:**

- W1: The most significant improvement is observed in the deep research benchmark using clarification data created by the authors (in-domain). In contrast, on out-of-domain benchmarks, the performance improvement is marginal compared to most training-free approaches.
- W2: The paper shows that SFT works better on offline testing but worse on online testing, which is possibly due to overfitting. The training duration for SFT is the same as RL (3 epochs), despite SFT generally converging faster. The paper does not mention an early-stopping strategy to prevent overfitting, which suggests the performance gap between SFT and RL might be overestimated.

---

> ### Author Rebuttal · Authors · 2026-03-31
>
> Thank you for your valuable suggestions! We provide the following responses to address your concerns point by point:
> > W1: Domain generalization
>
> The observed performance pattern stems from the specific task structures of out-of-domain benchmarks.
> - As noted in Section 4.2, Rigorous Bench tasks often demand up-to-date knowledge (e.g., analyses of recent model releases beyond typical knowledge cutoffs), making retrieval quality the primary bottleneck rather than intent alignment. Users cannot clarify what they do not know they need. Moreover, all proactive baselines share these marginal gains, indicating the performance ceiling is imposed by task characteristics, not IntentRL limitations.
> - In PDR-Bench, which evaluates personalization using authentic profiles and real-world tasks, we applied original task-user pairings without query simplification. Here, IntentRL shows significant gains on personalization dimensions (GOAL, CONT), directly reflecting strong intent exploration and confirming successful generalization beyond its training domain.
>
> Furthermore, IntentRL consistently outperforms training-free approaches (built-in Qwen and OpenAI clarify modules) on both benchmarks, increasing up to 15.31% on PDR-Bench. Compared to trained baselines (Learn-to-Ask, Tell Me More, CollabLLM), our agent requires only 50 seed training samples, minimizing resource costs while achieving state-of-the-art overall performance and demonstrating high efficiency.
>
> To further address your concern and validate domain generalization, we conduct an additional experiment. We train our agent exclusively on **Science & Technology** data from DR Bench and evaluated its simulated clarification through downstream report overall scores on three unseen domains: **Literature, Education & Jobs, and Health**.
> Method|Literature|Education & Jobs|Health
> -|-|-|-
> no clarify|34.82|42.85|45.02
> Qwen clarify|41.38|44.56|46.70
> CollabLLM|36.75|43.55|46.47
> IntentRL|47.34|47.41|49.51
>
> The results show that IntentRL yields substantial improvements over the "no clarify" baseline across all three unseen domains, significantly outperforming both the training-free (Qwen clarify) and the strongest trained (CollabLLM) baselines. This confirms that our framework effectively learns generalizable user-interaction behaviors that transfer robustly to entirely new domains, proving that our improvements in out-of-domain scenarios are highly effective and far from marginal.
> > W2: Overfitting of SFT
> > Q2: Training curves
>
> The 4-epoch (Appendix A.2) setting was not chosen arbitrarily but was determined after carefully analyzing empirical results. Due to the restrictions on updating images/links, we present the stage II SFT training curve of loss and the early-stop results on testing set in tabular form (use Qwen DR agent in DR Bench). As shown in the following table, we can see that the loss smoothly decreases and stabilizes in Epoch 4. Simultaneously, **clarification quality (Intent F1) and downstream report quality (Overall) improve continuously without degrading from epoch 1 to 4**. These results from both training and testing are inconsistent with overfitting. It fundamentally results from the inherent exposure bias caused by SFT training, which struggles to generalize to real-time interactive dynamics compared to our RL formulation, as we discuss in Section 4.4.
> Step|Loss
> -|-
> 10|0.9137
> 40|0.1439
> 67 (epoch 1)|0.1090
> 100|0.0592
> 135 (epoch 2)|0.0472
> 175|0.0345
> 202 (epoch 3)|0.0335
> 245|0.0273
> 270 (epoch 4)|0.0261
>
> Table 1. Training Loss at multiple steps
>
> Epoch|Intent F1|Overall
> -|-|-
> 1|15.33|40.24
> 2|16.68|40.89
> 3|17.65|41.07
> 4|18.75|41.07
>
> Table 2. Early-stop results per-epoch
>
> > Q1: Stage 2 SFT explanation
>
> As described in Section 3.3, we traverse the C-DAG to generate intent trajectories. In Stage I SFT, they serve as expert trajectories, teaching the agent to clarify intents sequentially.  For the Stage II online strategy in SFT, we also apply a partial teacher-forcing mechanism. Specifically, we seed the context with groundtruth prefixes of varying lengths from the training split and let the Stage I trained agent generate the next clarification.
>
> Formally, given a simple query $q_s$ and an expert intent trajectory $\mathcal{I}_{sft} = \lbrace I_1, I_2, \dots, I_k \rbrace$, the initial observation is $H_0 = \lbrace q_s\rbrace$. The agent generates a clarification $x_1$, and receives a simulated user response $u_1$.
> - On Hit: If $x_1$ captures the target intent $I_1$, the history updates to $H_1 = \lbrace q_s, x_1, u_1\rbrace$, and the target label advances to $I_2$.
> - On Miss: If $x_1$ misses $I_1$, the history still updates to $H_1 = \lbrace q_s, x_1, u_1\rbrace$, but the label remains $I_1$ to compel the agent to re-attempt the missed intent.
>
> These new input-label pairs are added to the Stage II training data, enabling agent to learn recovery behaviors from one-turn simulated trajectories even in SFT. We will elaborate on it in the next version.

---

> > ### Author Rebuttal · Reviewer_KMHY · 2026-04-03
> >
> > The rebuttal adequately addresses my concerns. I will raise my score to 4.

---

### Official Review · Reviewer_n6zx · 2026-03-13

**Soundness:** 3
**Presentation:** 3
**Significance:** 3
**Originality:** 3
**Overall Recommendation:** 5
**Confidence:** 3

**Summary:**

This paper argues that deep research agents should not immediately start searching from an underspecified user query, but should first ask clarification questions to infer the user’s latent intent. To do this, the authors propose IntentRL, which models clarification as a POMDP and trains a questioning policy with a two-stage RL framework built on a synthetic Clarification DAG dataset and simulated user rollouts. The reward encourages relevant, concise, and non-redundant questions. Experiments show that IntentRL asks better clarification questions than prior methods and that its refined queries improve the quality of downstream research reports, especially in comprehensiveness, depth, and instruction-following.

**Compliance With Llm Reviewing Policy:**

Affirmed.

**Final Justification:**

My main concerns are addressed, so I raise my score.

**Key Questions For Authors:**

1. The appendix provides a case study of the proposed method, which is helpful. Could the authors further provide more qualitative analysis comparing different methods? For example, it would be useful to see side-by-side clarification trajectories or downstream outputs from IntentRL and strong baselines, so that readers can better understand where the gains actually come from, what kinds of intents are captured better, and in what cases the methods behave differently.
2. The current setup relies substantially on synthetic intent structures and simulated interactions. Could the authors discuss more explicitly how they expect the method to generalize to real users, whose preferences may be noisier, less consistent, or not cleanly representable as a predefined intent set?

**Limitations:**

The paper does not include a paragraph in the main text that discusses its limitations in detail. The authors may discuss more about the limitations of their method in real-world applications.

**Strengths And Weaknesses:**

**Soundness.** The paper is technically meaningful and presents a fairly complete RL training pipeline for proactive clarification in deep research. The overall formulation is reasonable, and the empirical results are generally consistent with the paper’s main claim that better clarification can improve downstream performance. My main concern is that the training setup still relies heavily on synthetic structure, which makes it hard to judge how well the learned policy would transfer to messier real-world user intents. In addition, while the method does improve downstream DR performance, those gains often appear somewhat marginal, so it is unclear whether the data augmentation and training cost is always justified.

**Presentation.** The paper is generally well written and easy to follow. The motivation is clear, the high-level story is coherent, and the method is organized in a way that makes the overall pipeline understandable despite having several components. I also think the paper does a good job connecting the problem setting, the proposed method, and the empirical evaluation.

**Significance.** The paper addresses an important problem for long-horizon research agents, since clarifying user intent before launching an expensive research process is clearly valuable in practice. I also find the setup timely and relevant as deep research systems become more capable. That said, the current evidence feels somewhat benchmark- and pipeline-dependent, and the reward design depends on a relatively explicit target intent set. This makes me less certain about how broadly the approach would generalize beyond settings where user intent can be cleanly represented in this way.

**Originality.** I think the paper has a reasonable level of originality overall. The individual ingredients are not radically new on their own, but the paper combines them in a fairly natural and non-trivial way for the specific problem of proactive clarification in deep research.

---

> ### Author Rebuttal · Authors · 2026-03-31
>
> We thank the reviewer for your valuable suggestions and address concerns point-by-point:
> > Concern 1: Generalizability to real-world scenarios
>
> Regarding the transition from synthetic C-DAG to noisy real-world preferences, our two-stage RL framework bridges this gap. While Stage I utilizes offline expert trajectories from C-DAG, Stage II explicitly mitigates offline distribution shift via an online, intent-aware user simulator. The clarification generation score in Table 3 shows that Stage II reinforces adaptive responses across diverse, unpredictable user reactions.
>
> To prove generalization to messy real-world intents, we deliberately evaluated IntentRL on PDR-Bench. PDR-Bench consists of 50 personalized instances utilizing 25 authentic, complex user profiles instead of synthetic rubric-derived intents. Evaluated without query simplification, IntentRL outperformed all baselines, particularly on personalization dimensions (GOAL, CONT). We also conduct an additional experiment to validate out-of-domain generalization in our response to reviewer KMHY's W1. **These sufficient evidence prove our learned policy is robust and does not overfit to cleanly predefined intent lists.**
>
> Furthermore, our qualitative case study in Appendix C demonstrates the agent dynamically pivoting its focus to explore alternative long-term effects when a user explicitly rejects a clarification. This responsiveness confirms its capability to manage the messy realities of human interaction.
>
> The challenge of generalizing to real users is a shared limitation in simulation-based RL (e.g., UserRL, Qian et al.), not unique to our work. We acknowledge this and argue that the current evidence provides a principled, empirically grounded basis for confidence in real-world applicability, with full real-user deployment left as future work. A detailed discussion will be added to the Limitation Section.
> > Concern 2: Gain-cost imbalance
>
> The performance gains are jointly influenced by downstream DR agent capability and benchmark task structures.
> - DR agents with stronger search and reasoning like Gemini benefit more from IntentRL. On DR Bench, IntentRL improves the overall score by 14.89% over the no-clarify baseline.
> - Marginal gains on Rigorous Bench stem from its tasks demanding up-to-date knowledge (e.g., analyses of recent model releases beyond typical knowledge cutoffs), where the bottleneck is retrieval quality, not intent alignment (as noted in Section 4.2). Users can't clarify what they do not know they need. And this ceiling affects all proactive baselines equally, as shown in Table 2.
> - Conversely, on PDR-Bench, IntentRL shows significant gains in key personalization dimensions (GOAL, CONT), proving its strong intent exploration capability.
>
> Regarding costs, IntentRL requires only 50 seed training samples, and fine-tuning a 7B model for 4 epochs is a one-time upfront investment. This cost must be weighed against the alternative: **executing an ambiguous query without clarification often results in massive computational resources and time during long-horizon DR tasks**. Generating a few clarification turns is computationally negligible compared to the massive resources wasted when a heavy DR agent pursues research in the wrong direction. The cost-benefit trade-off strongly favors proactive clarification.
>
> > Concern 3: Qualitative analysis of baselines
>
> As external images/links are restricted here, we provide a textual side-by-side analysis based on the case "Discuss the influence..." in Figure 5. This case pairs a target set of 14 intents (denoted as A-N). User feedback to irrelevant and repetitive questions is denoted as R and P.
> - Tell Me More: immediately outputs "I have understood what you mean, and we can proceed with the investigation", terminating clarification without hitting any intent. The downstream report covers 2 related rubrics (Overall: 43.33). This shows SFT-only models struggle with sustained exploration in new domains.
> - Learn to Ask: asks about positive vs. negative impacts → hits G → asks about specific case studies → user replies R → rephrases the same question → user replies P. It hits 3 intents and 6 related rubrics (Overall: 45.76), demonstrating that offline-only training lacks sensitivity to live feedback.
> - CollabLLM: ask about personal & professional relationships → hits D → ask about  methodologies  → user replies R → ask about improvement and potential risks → hits G → ask about near-term & long-term effect → hits I → ...  It hits 5 intents and 8 related rubrics (Overall: 46.95).
> - IntentRL: The trajectory is depicted in Figure 5. Our method robustly adapts to feedback, hitting 7 intents and 9 related rubrics (Overall: 47.41).
>
> **Conclusion:** Capturing more intents during clarification directly enriches downstream reports, driving higher rubric hits and overall scores. Besides, comparative intents (e.g., "A vs. B") are captured better. We will add this qualitative comparison to the revised appendix.

---

> > ### Author Rebuttal · Reviewer_n6zx · 2026-04-04
> >
> > Thank the authors for their rebuttal, and I have raised my score.

---

### Decision · Program_Chairs · 2026-04-30

**Decision:**

Accept (regular)

**Comment:**

This paper addresses the problem of ambiguous user intent in deep research, and propose IntentRL, a framework to train proactive agents that clarifies with user intents.

Strength:
* The paper is written clearly with a strong motivation.
* The framework achieves good results.

Weakness:
* Reviewers questioned about the generalizability of the framework in the real-world. The discussion in the rebuttal generally makes sense – it'd be good to include them in the camera-ready version and acknowledge the limitation.
* I agree with the reviewer that the fidelity of the user simulation could be one concern, and it'd be better to include real human studies or at least acknowledge it as a limitation as well.